# Formation of supramolecular channels by reversible unwinding-rewinding of bis(indole) double helix via ion coordination

Debashis Mondal [1], Manzoor Ahmad [1], Bijoy Dey [2], Abhishek Mondal [1] & Pinaki Talukdar [1] ✉

Stimulus-responsive reversible transformation between two structural conformers is an essential process in many biological systems. An example of such a process is the conversion of amyloid-β peptide into β-sheet-rich oligomers, which leads to the accumulation of insoluble amyloid in the brain, in Alzheimer's disease. To reverse this unique structural shift and prevent amyloid accumulation, β-sheet breakers are used. Herein, we report a series of bis(indole)-based biofunctional molecules, which form a stable double helix structure in the solid and solution state. In presence of chloride anion, the double helical structure unwinds to form an anion-coordinated supramolecular polymeric channel, which in turn rewinds upon the addition of Ag⁺ salts. Moreover, the formation of the anion-induced supramolecular ion channel results in efficient ion transport across lipid bilayer membranes with excellent chloride selectivity. This work demonstrates anion-cation-assisted stimulus-responsive unwinding and rewinding of artificial double-helix systems, paving way for smart materials with better biomedical applications.

The reversible transformation between two distinct self-assembled forms, aided by conformational changes, is ubiquitous in various biopolymers, including oligonucleotides and peptides[1–5]. Such a dynamic structural change from one state to the other is usually triggered by external stimuli, e.g., H⁺ [6,7], anions[2,8,9], cations[10], saccharides[11], light[12–15], etc. The interplay of noncovalent forces such as hydrogen bonding[16–19], electrostatic interaction[20], π–π stacking[21–23], van der Waals interactions, etc.[24,25], ascertains the relative stability of these self-assembled forms. One such example of chemical transformation via conformational change is seen in the conversion of amyloid-β peptide structure into β-sheet-rich oligomeric forms in Alzheimer's disease. The process leads to an accumulation of insoluble amyloid deposits in the brain. In order to stop amyloid deposition, β-sheet breakers are used as drugs that specifically bind to the amyloid-β peptide, thereby blocking or reversing this unusual conformational change[26]. Another example is the spontaneous folding of human telomeric DNA into a

secondary G-quadruplex structure in the presence of K⁺ ion that occurs through conformational changes in the DNA structure[27–29]. The reverse process, which is essential for transcription, is assisted by the hPOT1 protein[30]. Conformational change-assisted structural transformation can also be seen during the transcription process, where the unwinding of a DNA double helix occurs in presence of helicase enzymes[31–33]. Significant efforts have been made to introduce synthetic supramolecular systems that can interchange between two structurally different forms in response to external stimuli. For example, a silver ion complex of a conformationally flexible bipyridine molecule, described by Lee and co-workers, can form different self-aggregated secondary structures depending on the size of its counter anion[2,34]. In another report, Inouye and co-workers described the unfolded form of an oligo(m-ethynylpyridine) polymer, which changes into a well-ordered folded conformation upon interaction with saccharides[11]. Along with switchable secondary supramolecular structures, the interconversion

[1]Department of Chemistry, Indian Institute of Science Education and Research Pune, Dr. Homi Bhabha Road, Pashan, Pune 411008 Maharashtra, India. [2]Department of Chemistry, Tata Institute of Fundamental Research, 36/P, Gopanpally Village, Hyderabad 500046 Telangana, India. ✉ e-mail: ptalukdar@iiserpune.ac.in

of helical assemblies (e.g., single helix, double helix, triple helix, anion helicate, etc.) driven by external stimuli has also contributed to the field significantly. For example, Flood and co-workers reported triazole-based anion helicates that undergo a conformational switching between single helix to double helix upon treatment with anion or other external stimuli[35,36]. The work on switchable anion helicates is thoroughly explored in the seminal works of Jeong[9,37–41], Wu[42–44], Berryman[45,46], Yashima[47–49], and co-workers, which report stimuli-induced helical assembly formation and their switchable behavior. However, artificial systems that can closely mimic natural congeners, like the unwinding of DNA double helix into open conformation, are rare, and only a few examples have been reported in the literature. For example, Yashima and co-workers reported an oligoresorcinol-based double helix, which tranforms into its unwind form in the presence of cyclodextrin[50]. Due to the lack of sophisticated and complex building blocks that can form stable dual-faced aggregates, such synthetic system are rare[11,51–53].

This work demonstrates a reversible switching of bis(indole)-based isophthalimides **1a−1d** between two distinct self-assembled conformations. The compounds possess multiple hydrogen bond donor and acceptor sites, including the acidic N−H and C−H bonds of the indole moiety. With the help of these bonding sites, the bis-indole molecules form a self-assembled double-helical conformation by employing intermolecular hydrogen bonding interactions. Upon treatment with Cl⁻ ion, the double helix structures transform into an anion-coordinated polymeric conformation. In contrast to the reversible switching behavior of **1a−1d**, the **1e** derivative that lacks the C−H proton at the C-3 position of the indole moiety stays in its double helix conformation even after being treated with the chloride ion. In compounds **1c** and **1d**, an octyloxy chain was introduced into the central aromatic ring of the bis(indole) moiety to modulate their solubility in organic solvents. For compounds **1a** and **1d**, the trifluoromethyl group was attached at the para position of the terminal aromatic ring to enhance the transmembrane ion transport.

## Results and discussion
### Synthesis of bis-indole derivatives
The bis(indole) molecules **1a−1d** were synthesized starting from ethyl 7-nitro-1$H$-indole-2-carboxylate **6** (Supplementary Figs. 2, 3, 4, and 5). At first, the indole acid **7** was synthesized in four steps starting from 2-nitroaniline **2** following the reported literature procedure (Supplementary Fig. 1)[54]. Subsequently, compound **7** was reacted with different *para*-substituted aryl amines **8a−8b** in presence of EDC·HCl and HOBt to furnish the indole-amides **9a−9b** in 62−96% yield. Subsequent hydrogenation reaction of indole-amides **9a−9b** gave the reduced products **10a−10b** in quantitative yield. The amine derivatives **10a−10b** were eventually coupled with freshly prepared isophthaloyl dichlorides[55,56] **12a−12b** to furnish the desired compounds **1a−1d** in 82−86% yield. In order to synthesize the **1e** derivative, the indole ester compound **6** was first converted into the iodo-indole ester derivative **13** through an iodination reaction, and subsequently coupled with 4-*tert*-butylphenyl boronic acid through the Suzuki coupling to furnish 4-tert-butylphenyl substituted indole ester **14**. Following the above-mentioned synthetic procedures of **1a−1d**, the ester derivative **14** yielded the bis(indole) derivative **1e** (Supplementary Fig. 6).

### Double helix conformation
Single crystal X-ray analyses of bis(indole) compounds **1a** and **1b** were carried out to determine their solid-state structures (Supplementary Table 1). For compound **1a**, suitable crystals for X-ray analysis were obtained by slowly evaporating a 3:2 ethyl acetate-hexane solution of the compound. In the solid state, the bis(indole) molecule **1a** adapts a folded helical conformation. In this conformation, each carbonyl group attached to the indole moiety of one molecule participates in a bifurcated intermolecular hydrogen bonding interaction with the

indole N−H (H$_g$) and amide N−H (H$_c$) protons of the other molecule. There are four sets of bifurcated intermolecular hydrogen bonding interactions (Supplementary Table 2) between two helical units of **1a**, which leads to the formation of a double helix structure [**1a···1a**] (Fig. 1b). The hydrogen bond distances between N−H and carbonyl ranges from 2.84 Å to 3.18 Å. Along with these interactions, the double helix structure is also stabilized by π–π interactions between two helical strands with centroid-to-centroid interaction distances ranging from 3.59 Å to 4.18 Å. Furthermore, each double helix set is connected to its neighboring double helices by inter-helical hydrogen bonding interactions, through terminal amide N−H proton (H$_i$, facing outside) of one helix with the carbonyl group of the isophthalamide unit from the other helix. For compound **1b**, suitable crystals were obtained from 1:10 nitrobenzene-tetrahydrofuran solution. The X-ray analysis of compound **1b** shows a similar intermolecular hydrogen bonding assisted double-helix structure in the solid state (Fig. 1c).

The formation and stability of double-helical conformation in the solution phase were studied using various experimental techniques. The formation of dimeric structures by **1a** and **1b** in the solution phase was confirmed by the ESI-MS experiment. Initially, a solution of **1a** and **1b** (10 μM) were prepared separately in a solution of CH$_2$Cl$_2$:CH$_3$CN (10:1) and used for electrospray ionization mass spectrometric (ESI-MS) studies. The signal corresponding to [2M + H⁺] from the ESI-MS experiment (where M is the exact mass of **1a** or **1b**) indicates the formation of a dimeric structure in the solution phase (Supplementary Figs. 10 and 11). Further, to confirm the formation of a hetero-dimeric helical structure, ESI-MS experiment was performed for a mixture of compounds **1a** and **1b**. The mass spectroscopic data provided the signal at $m/z = 1429.4482$, which corresponds to the [**1a** + **1b** + H⁺] complex in the solution state and hence provides direct evidence for the formation of the hetero-dimeric structure (Supplementary Fig. 12).

The evidence in favor of the formation of double helical structure was obtained through the ¹H NMR spectroscopic studies of the mixture of compounds **1c** and **1d**. In this study, the alkyl derivatives **1c** and **1d** were used due to their better solubility in CDCl$_3$ (a non-hydrogen acceptor NMR solvent). The ¹H NMR spectrum of a mixture of **1c** and **1d** showed a new set of signals in addition to the NMR signals of the pure compounds. These new NMR signals confirms the formation of a hetero-dimeric aggregated structure in the solution phase (Supplementary Fig. 15). The claim of double helix formation was further supported by NOSEY NMR spectroscopic studies of the mixture of compounds **1c** and **1d**. The NOESY spectrum clearly shows the interactions between the H$_g$-H$_{g'}$ protons (Fig. 1d and Supplementary Fig. 16). These interactions are possible only if **1c** and **1d** helices come closer to form a mixed double helix structure (see the X-ray structures of the double helix) and therefore, confirmed the formation of the double helix structure in the solution phase.

Further, to check the stability of the double helix structure, a concentration-dependent ¹H NMR study of compound **1c** was performed in CDCl$_3$. The concentration variation of **1c** from 30.0 mM to 0.06 mM did not show any significant change in the chemical shift of indole and amide N−H protons (i.e., H$_c$ and H$_g$). This result suggests that the double helical conformation is stable even at a very low concentration (Fig. 1e). To know the nature of the hydrogen bond for the formation of the double helix structure, i.e., intramolecular or intermolecular, the DMSO-$d_6$ titration experiment was performed. Addition of DMSO-$d_6$, a hydrogen bond acceptor solvent, to a 5.0 mM solution of **1c** in CDCl$_3$ showed no significant changes in the chemical shift of indole and amide N−H protons, i.e., H$_c$ and H$_g$ (Fig. 1f). This result indicates the involvement of intermolecular hydrogen bonding interaction in the formation of a double helix structure. The chemical shift change (Δδ = 0.84) of terminal N−H proton (H$_i$) was observed because it faces outwards and is therefore, exposed to the hydrogen bond acceptor solvent. Moreover, we also performed ¹H NMR experiments of **1c** (5 mM) with the addition of urea (25 mM)[57] and cyanuric acid

(25 mM), which were expected to break the hydrogen-bonded double-helical structure. However, no significant changes in the chemical shifts were observed for acidic protons, except the $H_i$ proton (Supplementary Fig. 20b). The above results indicate the involvement of intermolecular hydrogen bonding interactions in the formations of the double helix structure. The stability of the double helix structure of **1c** (5 mM) was further examined at elevated temperatures (up to 100 °C) in $C_2D_2Cl_4$. This experiment resulted in a negligible change in the chemical shifts of N–H protons, thus highlighting the thermal stability of the dimeric structure (Supplementary Fig. 19). However, at a lower concentration of 0.1 mM, heating (up to 100 °C) causes a significant shift in the N–H proton resonance (Fig. 1g), indicating dissociation of the double-helical structure[16].

## Supramolecular anionic polymer formation

Having confirmed the double-helical structures of bis(indole) compounds in solid and solution states, the effect of chloride ions was investigated. The solid-state X-ray crystallographic analysis of **1b** with the chloride ion revealed the formation of the chloride-based anionic polymeric structure. To obtain suitable crystals for X-ray diffraction analysis, a mixture of **1b** and tetrabutylammonium chloride (TBACl) was slowly evaporated in acetonitrile. The addition of TBACl to the receptor solution drives the conformational changes across the C (indole aryl) and N (attached to $H_c$) single bonds (Fig. 2a) and

eventually leads to the binding through different hydrogen bonding interactions (Supplementary Table 5). Conformational changes in the receptor molecule creates three anion-binding sites. The central part of the receptor binds with the $Cl^-$ ion through hydrogen bonding interactions involving $H_b$, $H_c$, and $H_d$ protons, respectively ($d_{C \text{ or } N \cdots Cl} = 3.20 \text{ Å} - 3.55 \text{ Å}$). Whereas the terminal binding sites of the receptor are involved in connecting the neighboring molecules via $Cl^-$ ion bridging through hydrogen bonding interaction with $H_h$ ($d_{C \cdots Cl} = 3.46 \text{ Å}$), $H_i$ ($d_{N \cdots Cl} = 3.29 \text{ Å}$), and $H_j$ ($d_{C \cdots Cl} = 3.68 \text{ Å}$) protons, which eventually forms a linear polymeric chain. Besides the amide N–H proton ($H_i$), the C–H proton ($H_h$) also plays a crucial role in forming a hydrogen bond with the $Cl^-$ ion for the conformational change. The linear chains are connected by hydrogen bonding interaction between the terminal carbonyl of the receptor from one linear chain and the terminal aryl proton of the receptor of the other chain ($d_{C \cdots O} = 3.22 \text{ Å} - 3.26 \text{ Å}$). This along with other noncovalent interactions make a two-dimensional layer (Fig. 2a b). Eventually, these layers aggregate to form microsheet structures (Fig. 2a).

## Solid state morphology study

To get insights into the effect of $Cl^-$ ions on the self-assembly pattern of bis(indole) molecule **1b** through $Cl^-$ ion binding induced conformational changes, morphological studies using field emission scanning electron microscopy (FESEM), atomic force microscopy

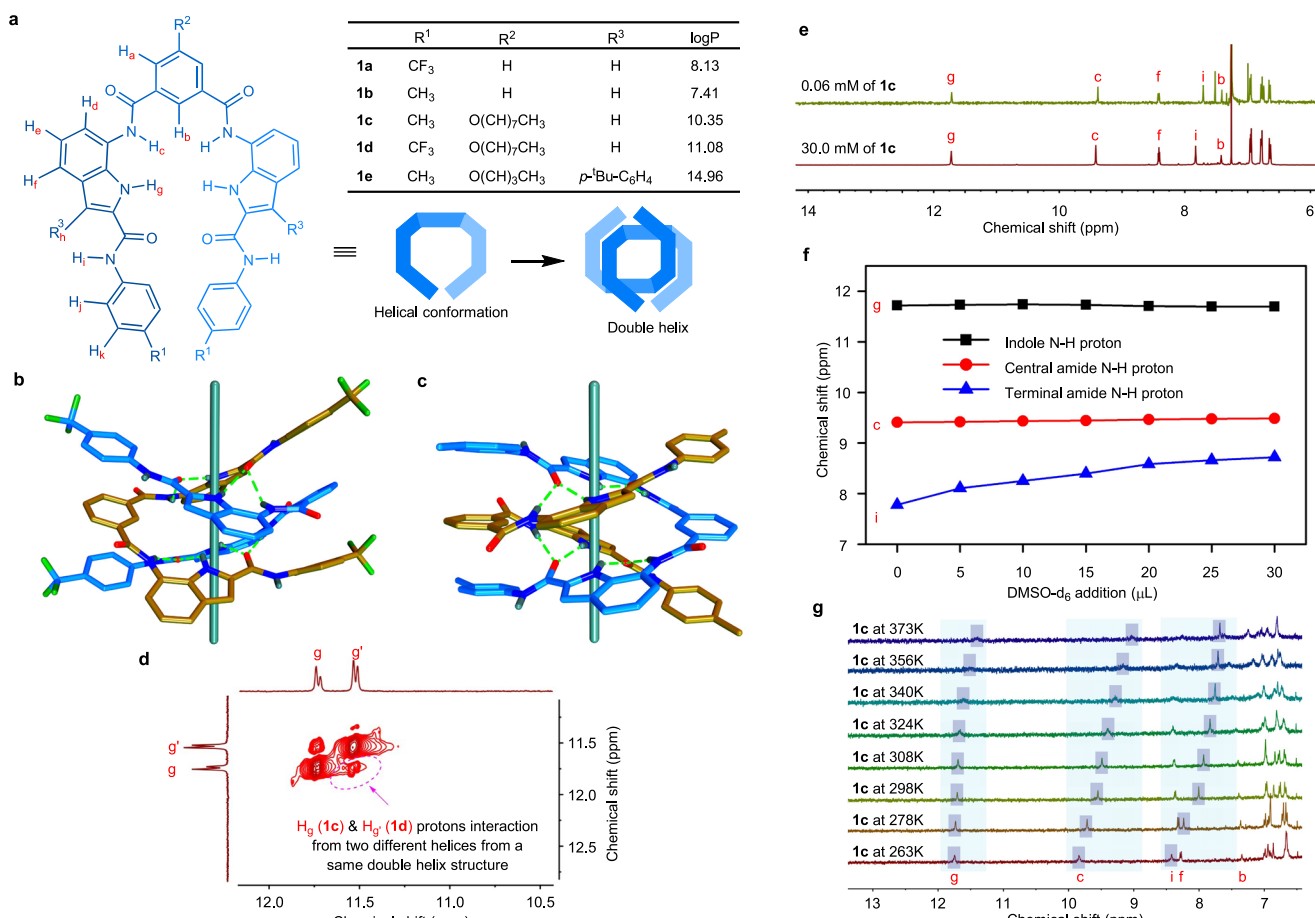

**Fig. 1 | Formation of double helix by Bis(indole) compounds.** Chemical structures of bis(indole) molecules **1a–1e** with single and double-stranded helical schematic representation (**a**). The side view of the X-ray crystal structures of **1a** (**b**) and **1b** (**c**) forming double helix structures. Two helices in each double helix structure are shown in two different colors, where hydrogen bonds are shown as dotted lines, and the cyan rods along the helical axis are added for a better understanding of the double helix. The solvent molecules in the crystal structure of

**1a** and **1b** were omitted for clarity. The partial ¹H–¹H NOESY NMR spectrum of a mixture of **1c** (3.0 mM) and **1d** (3.0 mM) in CDCl₃ showing the $H_g$ and $H_{g'}$ interaction from a hetero-dimeric double helix structure (**d**). The partially stacked spectra of ¹H NMR dilution experiment of **1c** in CDCl₃ at 25 °C (**e**). Chemical shifts of N–H protons in DMSO-$d_6$ titration of **1c** (5.0 mM) in CDCl₃ at 25 °C where the spectra were calibrated by tetramethylsilane (TMS) signal (**f**). The partially stacked ¹H NMR spectra of **1c** (0.1 mM) in $C_2D_2Cl_4$ recorded at variable temperature (**g**).

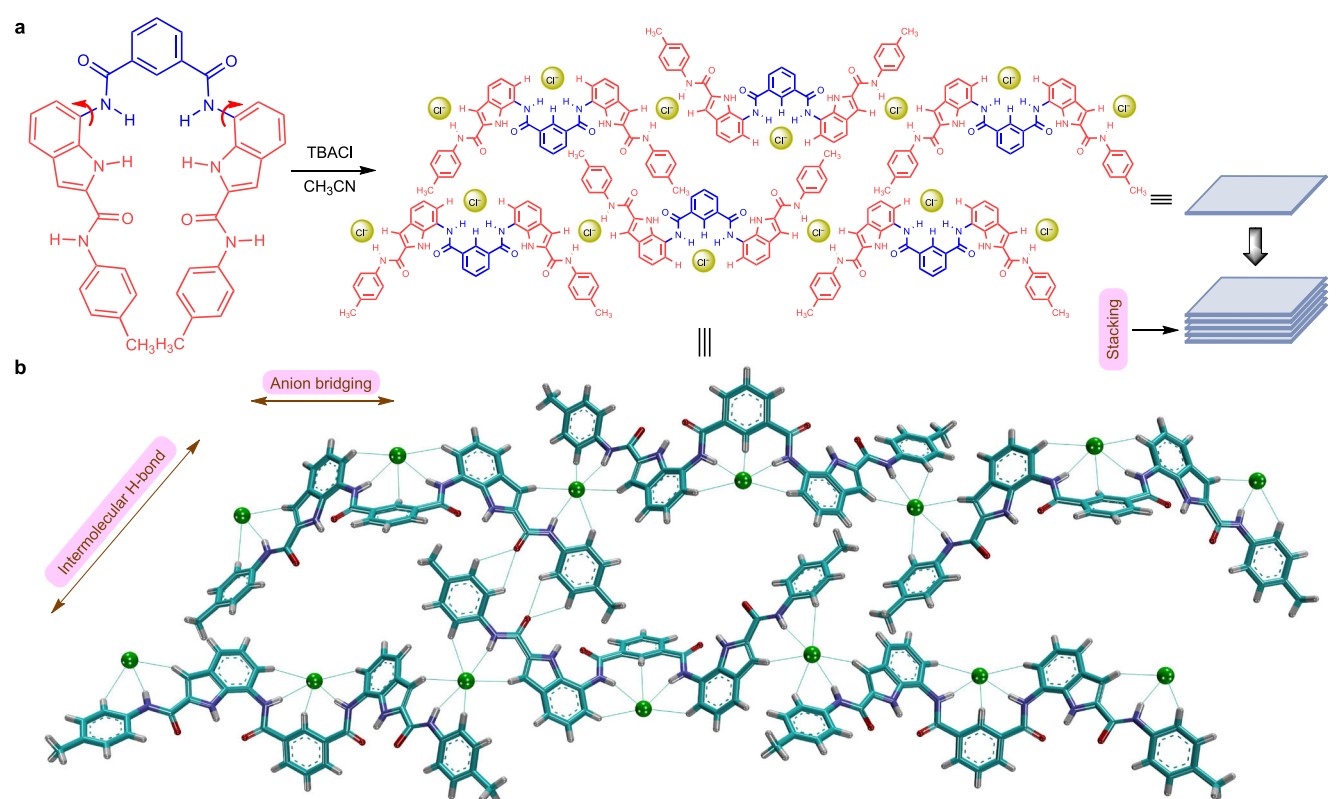

**Fig. 2 | Chloride-induced supramolecular polymer.** The schematic illustration of conformational changes across the single bond followed by the formation of anion-coordinated supramolecular assembly by **1b** (**a**). The top view of the formation of Cl⁻ ion coordinated self-assembled architecture from the crystal structure of **1b** with Cl⁻ (**b**). The tetrabutylammonium cations are omitted for clarity in the solid-state representation. The light blue dotted lines in the bottom crystal structure represent the hydrogen bonding interactions.

(AFM), and transmission electron microscopic (TEM) were performed. For these microscopy studies, 150 μM solution of **1b** and [**1b** + Cl⁻] in acetonitrile/THF (1:1) was drop-cast on a silicon wafer (or copper-grid for TEM) and air-dried. Initially, the morphology of the free receptor **1b** was investigated through FESEM analysis. The FESEM studies showed that the bis(indole) compound self-aggregates to form a globular structure with a diameter of 90–400 nm (Fig. 3a). The formation of globular self-assembly of **1b** was further confirmed by AFM imaging (Fig. 3c), where it has shown the curved height bar with a maximum height of 172 nm (Supplementary Fig. 23). The FESEM morphology of Cl⁻ ion-bound receptor complex showed multi-layered microsheets (Fig. 3b), formed by the aggregation of the anion-coordinated polymeric chain through different noncovalent interactions. The flat pattern of the height bar in the AFM image of the anionic complex (Supplementary Fig. 24), unlike the curved height bar observed for the free compound, indicates the formation of planar sheet-like aggregates. We also performed TEM analysis of **1b** and [**1b** + Cl⁻] complex to get a better understanding of their morphological behavior (Fig. 3e, f). The free compound was observed to form solid spherical aggregates with a similar diameter as seen in the FESEM studies. The globular structure of **1b** is expected to be the outcome of the accumulation of the non-planar double helices employing the inter-helical noncovalent interactions. On the other hand, the TEM image of the anionic complex shows the formation of bigger aggregates generated through the self-assembly of the multiple planar microsheets.

**Solution phase anion binding**

The formation of the Cl⁻ bound polymeric complex by unwinding the double helical structure of bis(indole) molecule **1b** in the solution phase was confirmed by ESI-MS studies (Supplementary Figs. 27 and 28) of a solution of **1b** (10 μM) with 3.0 equivalents of

tetramethylammonium chloride (TMACl) in CH₂Cl₂:CH₃OH (10:1 v/v). The data provided peaks corresponding to the different aggregated polymeric units (Supplementary Figs. 27, 28 and Supplementary Tables 6, 7), which supports the formation of a supramolecular polymeric complex in the solution phase. The formation of anion-induced supramolecular polymer in the solution phase was further validated by performing 2D-DOSY NMR experiments on the free and anionic complex of the receptor in CHCl₃ solution[58,59]. The 2D-DOSY NMR spectrum of the 5.0 mM solution of the free bis(indole) compound **1c** provided the diffusion coefficient value of 5.96 (±0.03) × 10⁻⁶ cm²/s (Supplementary Fig. 34). However, addition of 3.0 equivalents of TBACl salt to the receptor solution leads to a decrease in the diffusion coefficient value to 4.25 (±0.02) × 10⁻⁶ cm²/s (Supplementary Fig. 35). This decrease supports the formation of polymeric structure following the conversion of the double helix to the monomeric unit and subsequent formation of polymeric aggregates in solution. Furthermore, 2D-DOSY NMR analysis of **1c** in presence of 3.0 equivalents of TBACl yields diffusion coefficient values of 3.14 (±0.02) × 10⁻⁶ cm²/s and 2.64 (±0.02) × 10⁻⁶ cm²/s, for 14 mM and 20 mM of **1c**, respectively. Overall, in the presence of Cl⁻ ion, the diffusion coefficient values decreased from 4.25 (±0.02) × 10⁻⁶ to 2.64 (±0.02) × 10⁻⁶ cm²/s with increasing concentration of bis(indole) compound **1c**, from 5 to 20 mM (Supplementary Figs. 36 and 37). These observations suggest that **1c** forms a supramolecular polymer in the presence of Cl⁻ ion in the solution phase.

We performed ¹H NMR titration experiment of **1c** with TBACl to see the unwinding of the double helix structure and subsequent formation of the anionic complex via conformational change. Successive addition of TBACl salt (0 to 40 equiv.) to the receptor solution (3.0 mM) in CDCl₃ leads to the disappearance of the double helix proton signal (red star in Fig. 3g) with the appearance of a new set of

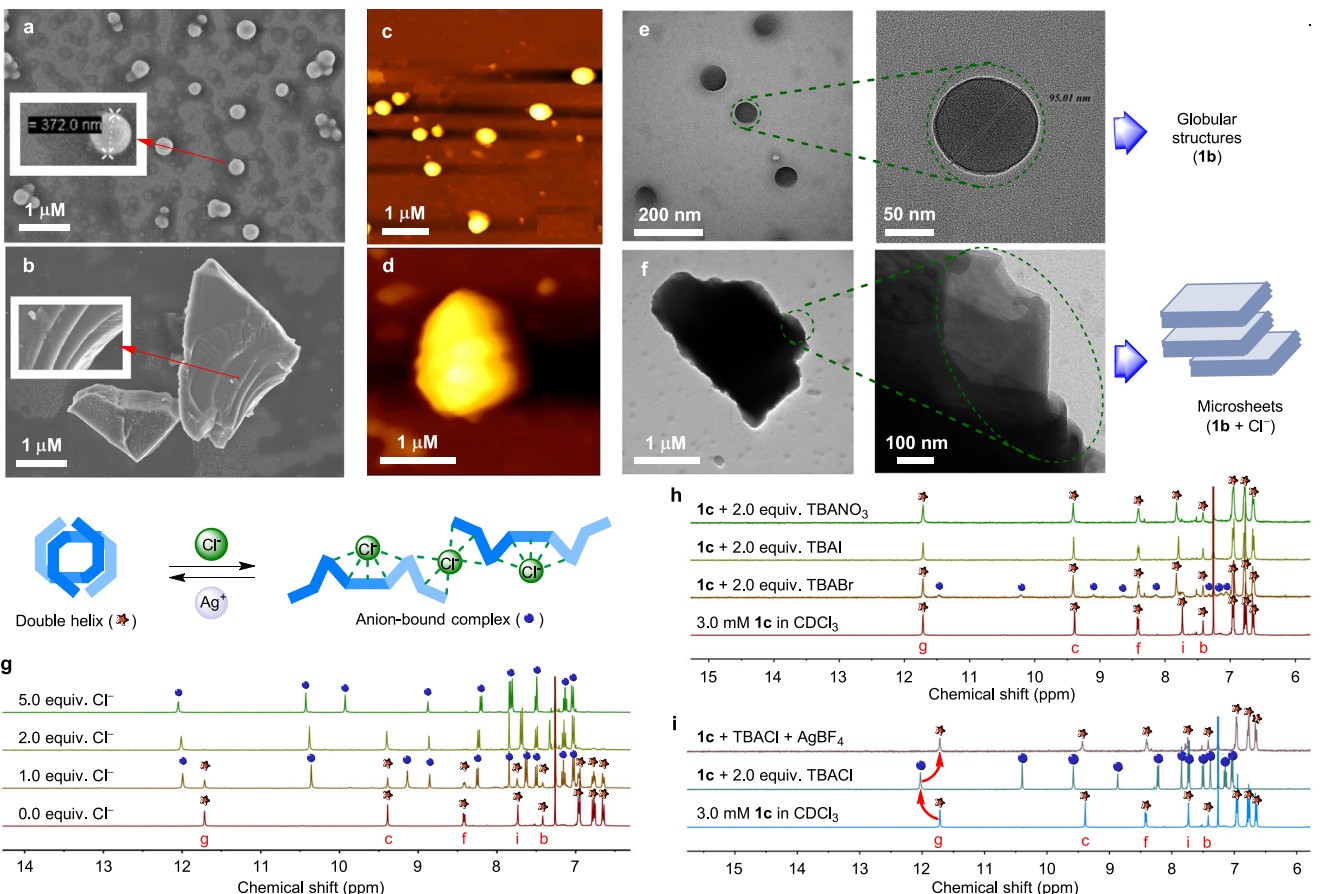

**Fig. 3 | Solid state morphology and solution phase anion binding studies.** The globular self-assembled structure observed in FESEM (**a**), AFM (**c**), and TEM (**e**) microscopic technics by **1b**. The multi-layer microsheets assembly captured in FESEM (**b**), AFM (**d**), and TEM (**f**) microscopic technics by **1b** with TBACl. The morphology studies were performed by preparing the 150 μM solutions of **1b** and [**1b**+Cl⁻] in acetonitrile/THF (1:1) solvent system. The stacked ¹H NMR spectra of **1c** (3.0 mM) in CDCl₃ at room temperature upon titrating with increasing equivalents of TBACl representing the conversion of the double helix to the anion bound complex (**g**). The stacked ¹H NMR spectra of compound **1c** in CDCl₃ upon the addition of 2 equivalents of TBABr, TBAI, and TBANO₃ at room temperature (**h**). The stacked ¹H NMR spectra of compound **1c** in CDCl₃ with the sequential addition of 2 equivalents of TBACl, and 2.5 equivalents of AgBF₄ to monitor the reversible unwinding-rewinding process (**i**).

proton signals (blue sphere in Fig. 3g). The appearance of a new set proton signals at a slightly downfield region is the outcome of receptor-anion interaction followed by the formation of an anionic complex. After each addition of TBACl, two distinct sets of proton signals corresponding to the free and anion-bound receptor complex can be seen because of a slow exchange between two conformations. However, upon the addition of 2.0 equivalents of TBACl to the receptor solution, the proton signal corresponding to the double helix compound completely disappears, suggesting the predominance of the anion-bound form. When similar anion binding studies were performed with 2.0 equivalents of TBABr, a much lesser conversion (10%) from dimer to anionic complex was observed (Fig. 3h). Contrarily, no accountable conversion and chemical shift change was observed when similar experiments were performed with iodide and nitrate salts. These observations indicate that the conformational change and the formation of the anionic complex are highly selective for Cl⁻ ion. Further, a series of ¹H NMR experiments with TBACl and silver salts were performed to check the reversibility of anion unwinding associated with conformational change. In this experiment, the addition of TBACl (2.0 equivalents) into a receptor solution of a double helix leads to a complete conversion into the polymeric complex. However, the addition of silver salts (2.5 equivalents of AgBF₄ and AgPF₆) to the above polymeric solution leads to the restoration of the double helix conformation (Fig. 3i). These observations indicate that the anion unwinding and rewinding

of double helical conformation are reversible and can be tuned by external stimuli such as Cl⁻ and Ag⁺ ions.

## Ion transport application of supramolecular polymeric system across phospholipid bilayer membrane

The formation of supramolecular polymeric chloride channel by bis(indole) molecule **1b** with Cl⁻ ion in solid and solution states (Fig. 4a) motivated us to evaluate their ion transport activity across a lipid bilayer membrane. For a thermodynamically stable ion channel, the interactions between the ion and channel-forming molecules have to be sufficiently strong to compensate for the dehydration energy of the ion[60]. The supramolecular ion channel formed through an ion-induced self-assembled polymeric structure can be a good option for providing a static hydrophilic pathway for the ions to pass through the hydrophobic lipid bilayer membrane[61–64]. For example, Matile and coworkers introduced the ion channel formation by dendritic folate molecule where hydrogen bond directed self-assembly of guanosine units form circular oligomers templated by a cation[61]. These circular oligomers finally form a supramolecular rosette channel for ion transport involving π−π interactions between the layers. Later, Davis and co-workers also reported the ion channel behavior by the cation-induced polymeric channel formed by nucleoside−sterol conjugate[62]. The guanosine moiety present in the molecule forms a G-quartet structure induced by the K⁺ ion, which subsequently stacks via π−π interaction to create a large and stable polymeric supramolecular

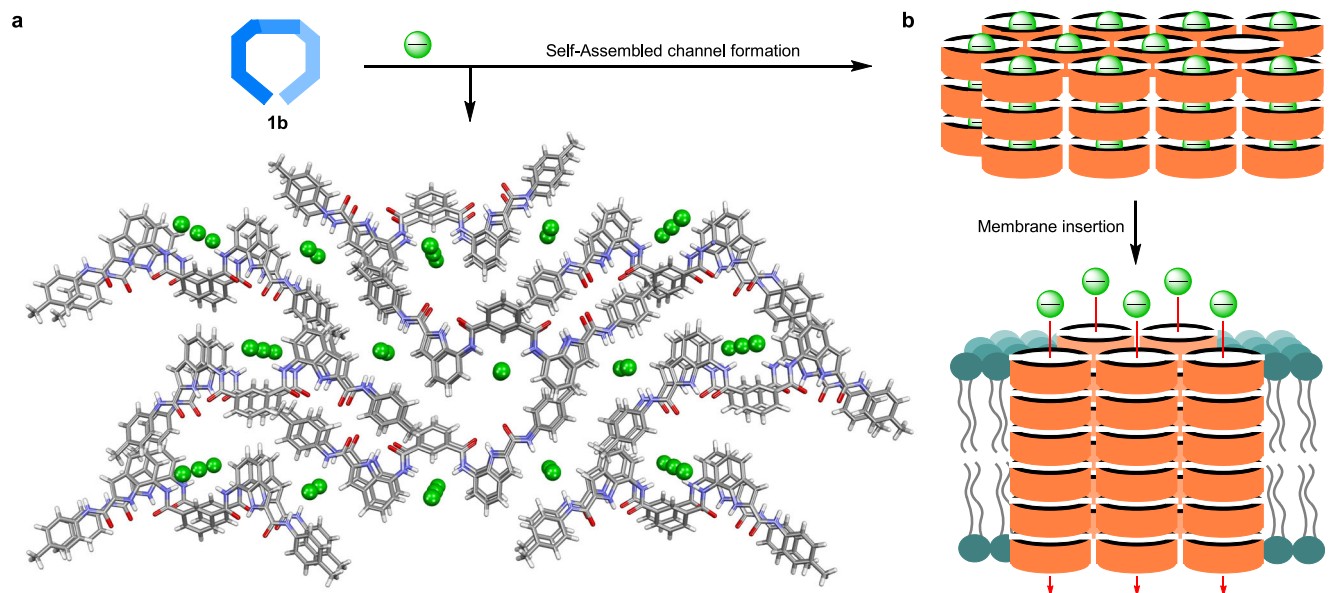

**Fig. 4 | Anion-induced supramolecular channel.** The crystal structure of the Cl⁻ coordinated supramolecular assembly of **1b** which stacked with a proper alignment to form an array of ion i.e., Cl⁻ channel (**a**). Here only three layers have been shown for stacking. Note that the bigger size tetrabutylammonium cations in aggregated structure partially block the channel path and hence remove for clarity. In addition, ion transport studies were investigated using **1b** in NaCl buffer solution where tetrabutylammonium cation was absent, and hence the chloride channel path will be free. The proposed schematic representation of the formation of chloride channel inside the lipid bilayer membrane from a Cl⁻ ion bound supramolecular assembly and then the transmembrane transport of Cl⁻ ion through this supramolecular channel (**b**).

structure for the transportation of ions through the lipid bilayer membrane. However, the anion-template-induced self-assembled supramolecular ion channels that can transport ions across the lipid bilayer membranes are extremely rare (Fig. 4b)[65].

At first, we evaluated the ion transport activity of the molecules **1a–1d** across the lipid bilayer membrane[66–68]. For that, vesicles entrapped with a pH-sensitive 8-hydroxypyrene-1,3,6-trisulfonate dye (HPTS) in 100 mM NaCl solution were prepared using egg yolk phosphatidylcholine (EYPC) lipids, and a pH gradient ($\Delta$pH = 0.8) was created between the intra- and the extravesicular medium[69–72]. The destruction of the pH gradient by transporter molecules was monitored by measuring the fluorescence intensity increment of the HPTS dye with time. Under a comparable concentration of 0.07 μM, the HPTS assay showed methyl derivative **1b** to be the most active compound (with 94% of fluorescent intensity after 200 s of transporter addition), followed by **1a** (84%) and **1c** (19%). The alkoxy chain derivative **1d** (12% at 0.07 μM) was found to be the least active among the four compounds (Fig. 5a). The above ion transport activity sequence of **1b** > **1a** > **1c** > **1d** is the outcome of the lipophilicity of the transporter molecules, where the compounds with a larger deviation of the logP values from 5 showed a lower ion transport activity. This explains why the octyl chain-bearing **1d** with a logP value of 11.08 was the least active among the four molecules[73]. Subsequently, concentration-dependent ion transport activities for compounds **1a–1c** were evaluated across the HPTS vesicles. The transport studies for **1d** were excluded due to its low activity compared to the other derivatives. The concentration-dependent activity of **1b** is shown in Fig. 5b. The Dose-dependent Hill analysis provides the $EC_{50}$ values of 13.7 nM, 10.9 nM, and 341.5 nM for compounds **1a**, **1b**, and **1c**, respectively (Supplementary Figs. 39, 40, and 41). Compound **1b** showed the highest activity with Hill coefficient value $n \approx 1$ (Supplementary Fig. 39). The Hill coefficient values $n$ close to one indicate the formation of a thermodynamically stable ion channel across the bilayer membrane[74,75].

In the HPTS-based ion transport experiment, the transporter molecules were added to the vesicular solution containing 100 mM NaCl solution. We expected the Cl⁻ ions from the solution to promote a similar channel formation (like the crystal structure) inside the hydrophobic lipid bilayer membrane, which will eventually aid the transport of Cl⁻ ions across the bilayer membrane. We also performed ion transport experiment by mixing the transporter molecule with the NaCl salt prior to its addition to the vesicular solution (Supplementary Fig. 42). The addition of this chloride-bound transporter molecule does not make any significant difference in the ion transport activity as compared to the addition of the free transporter molecule. These observations suggested that the ion channel behavior remains the same using either the free transporter molecule **1b** or the Cl⁻ ion bound receptor complex.

To get further insights into the ion transport mechanism, the most active compound methyl bis(indole) derivative **1b** was used for studying the ion selectivity across the HPTS vesicles[76–79]. By changing the extravesicular cations (Li⁺, Na⁺, K⁺, Rb⁺, and Cs⁺) at a 12 nM transporter concentration of **1b**, no significant change in the ion transport activity was observed (Supplementary Fig. 43), which rules out the role of cations in the ion transport process. This result excludes the possibility of M⁺/Cl⁻ symport and M⁺/H⁺ antiport mechanism out of four possible ion transport modes, viz., Cl⁻/OH⁻ antiport, H⁺/Cl⁻ symport, M⁺/Cl⁻ symport, and M⁺/H⁺ antiport. However, varying the extravesicular anions (Cl⁻, Br⁻, I⁻, and NO₃⁻) makes a significant change in the ion transport activity with pH dissipation trends in the order of Cl⁻ > NO₃⁻ > I⁻ > Br⁻ (Fig. 5c). This indicates the involvement of anions in the ion transport process with the possible ion transport modes of either Cl⁻/OH⁻ antiport or H⁺/Cl⁻ symport.

The influx of chloride ions for the most active compound **1b** was studied across the lucigenin (a chloride-sensitive dye)-based LUVs[80–82]. A Cl⁻/NO₃⁻ gradient was created across the vesicular membrane by adding NaCl solution to the external buffer. The influx of the Cl⁻ ions by the transporter molecule was monitored by measuring the rate of the fluorescence quenching of lucigenin dye at $\lambda_{em}$ = 535 nm ($\lambda_{ex}$ = 455 nm, for lucigenin). The Dose-responsive chloride transport studies for **1b** are shown in the Fig. 5d. The Hill analysis yielded the $EC_{50}$ value of 31.3 nM and $n$ value of 1, an indication of a stable supramolecular ion channel formation (Supplementary Fig. 46). Moreover, varying the

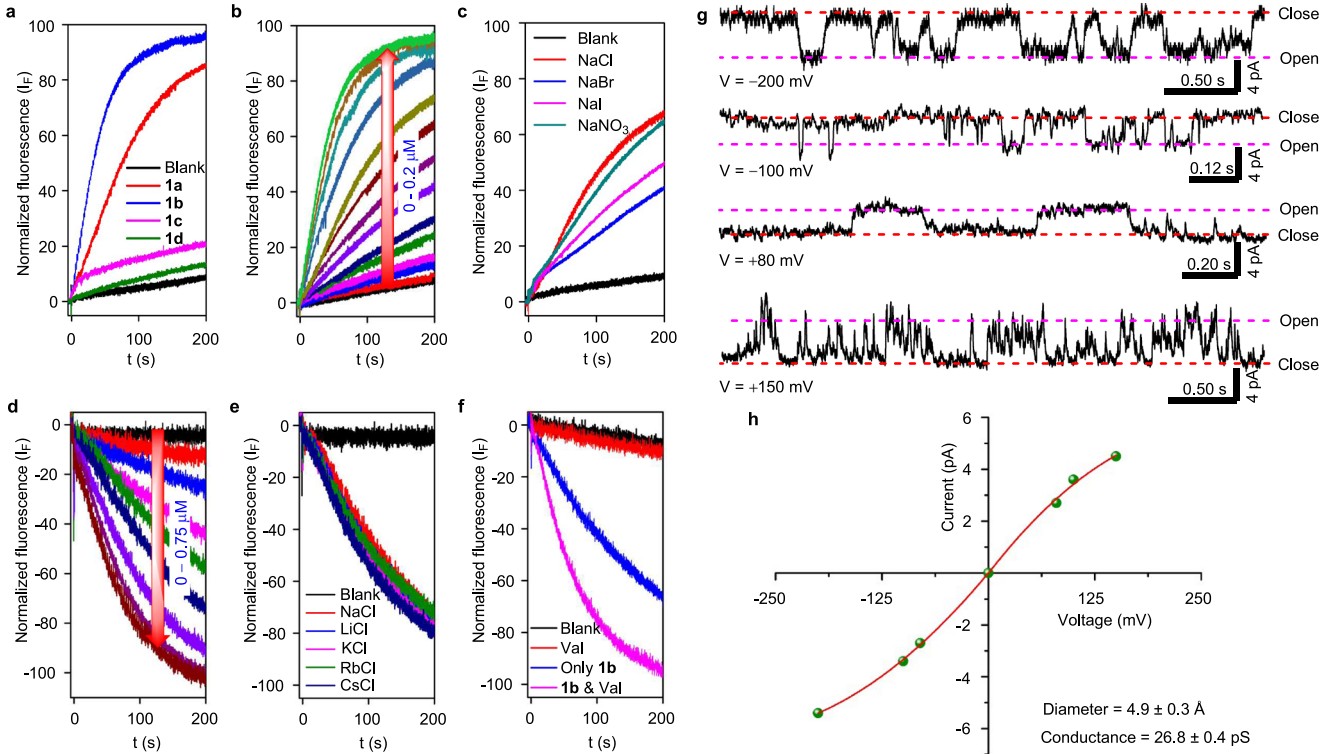

**Fig. 5 | Ion transport studies across lipid bilayer membranes.** The comparison of ion transport activities of **1a**–**1d** in the HPTS assay at 0.07 µM transporter concentration (**a**). The concentration-dependent ion transport activity of **1b** ($c = 0$–0.2 µM) in HPTS assay (**b**), and the ion transport activity of **1b** ($c = 12$ nM) in the HPTS assay by varying the extravesicular anions i.e., Cl⁻, Br⁻, I⁻, and NO₃⁻ (**c**). The concentration-dependent ion transport activity of **1b** ($c = 0$–0.75 µM) in the lucigenin assay (**d**). The ion transport activity of **1b** ($c = 0.12$ µM) in the lucigenin assay by varying the extravesicular cations i.e., Li⁺, Na⁺, K⁺, Rb⁺, and Cs⁺ (**e**). The ion transport activity of **1b** ($c = 0.10$ µM) in the lucigenin assay in the absence and the presence of valinomycin (0.5 µM) (**f**). The single channel current traces by **1b** ($c = 15$ µM) recorded at −200 mV, −100 mV, +80 mV, and +150 mV under symmetrical KCl solution (**g**). The plot current traces vs voltage obtained from the channel opening data at different potentials of **1b** fit in sigmoidal equation (**h**).

extravesicular cations (Li⁺, Na⁺, K⁺, Rb⁺, and Cs⁺) do not affect the transport activity (Fig. 5e), indicating no role of cations in the ion transport activity, thereby excluding the possibility of a symport mechanism. The only possible mechanism that remains for ion transport is therefore Cl⁻/X⁻ antiport. The direct confirmation of Cl⁻/X⁻ antiport mechanism by **1b** was obtained by valinomycin coupled lucigenin assay in KCl buffer. A multi-fold enhancement in the ion transport activity of transporter **1b** was observed in the presence of valinomycin, when compared to the transporter alone (Fig. 5f). In antiport mode, in presence of valinomycin and KCl, the influx of Cl⁻ ions can occur through (a) Cl⁻/NO₃⁻ antiport and (b) K⁺/Cl⁻ symport process by cooperative effect of transporter and valinomycin. However, the cooperative transport of K⁺/NO₃⁻ symport has a comparatively lesser effect, as the concentration of NO₃⁻ ions are the same in intravesicular and extravesicular solutions. The synergistic effect of valinomycin and the transporter enhanced the overall transport activity of the transporter in presence of valinomycin, compared to the transporter alone. On the other hand, in symport mechanism, a similar cooperative effect between the transporter and valinomycin will not occur as the transporter will be self-sufficient in transporting K⁺ along with Cl⁻, thereby maintaining electroneutrality. Therefore, the activity of the transporter molecule will be similar in the presence of valinomycin compared to the transporter alone.

The transport of ions through the formation of ion channel structures inside the lipid bilayer membranes by **1b** was examined by conductance measurement experiment with the help of planar lipid bilayer workstation[74,83–86]. In this experiment, a thin lipid bilayer membrane was constructed using diphytanoyl-phosphatidylcholine lipid (DPhPC) over a tiny aperture that separates the two chambers (*cis*

and *trans*) containing 1 M of KCl solution. The addition of **1b** in the *trans* chamber leads to the channel opening and closing behavior upon applying a potential gradient between two compartments, confirming the formation of ion channels inside the bilayer membrane. Furthermore, the channel opening and closing events were also observed at various negative and positive potentials, i.e., −200 mV, −100 mV, −80 mV, +150 mV, +100 mV, and +80 mV (Fig. 5g and Supplementary Fig. 49). Subsequently, the current traces obtained at different potentials were plotted against the corresponding voltage values, where the sigmoidal fit (Supplementary Fig. 50) indicates the Ohmic behavior by the channel (Fig. 5h). The conductance and diameter values were calculated using the Hille equation (Supplementary Equation 5) and were found to be $26.8 \pm 0.4$ pS and $4.9 \pm 0.3$ Å, respectively[72,77,87,88].

To validate the formation of an anion-induced supramolecular ion channel structures by the bis(indole) molecules inside the lipid bilayer membrane, we synthesized another derivative **1e**, where $H_h$ proton was replaced with the 4-(*tert*-butyl)phenyl group (Supplementary Fig. 6). As indicated by the crystal structure of the anionic complex of bis(indole) **1b**, $H_h$ proton plays a crucial role in the formation of anion-induced supramolecular polymeric structure. This interaction was further reflected in the ¹H NMR titration experiment of **1b** with TBACl, where significant chemical shift change was observed for the $H_h$ proton (Fig. 3g and Supplementary Fig. 29). However, when the similar ¹H NMR titration experiment of **1e** was carried out with TBACl (up to 5 equivalents), no chemical shift change was observed for any proton (Supplementary Fig. 32). This observation indicates that molecule **1e** exists as a double helix structure even in the presence of Cl⁻ ions, and no supramolecular polymeric structure, similar to **1b**, is formed. The

stability of **1e** double helix structure even in the presence of Cl⁻ indicates that the presence of a steric group at the H$_h$ proton position completely stops the formation of either chloride-bound supramolecular polymer (similar to **1b**) or any other chloride-bound complex (Supplementary Fig. 51). Therefore, the transport of Cl⁻ by the bis(indole) molecules via single-column channels or different unidentified assembly modes is unlikely.

In summary, we have designed and synthesized a family of bis(indole) based bio-functional molecules that adopt a double helical conformation in the free form, and forms anion-induced supramolecular ion channel structures in the presence of Cl⁻ ions. The systems demonstrate an efficient "anion-cation"-assisted stimulus-responsive unwinding and rewinding of the artificial double-helix systems. Crystallographic analysis confirms the double-helical conformation in the free form and supramolecular polymeric structure in its anion-bound form. The formation of different self-assembled structures in free and complex forms of **1b** was further supported by various morphological analyses using FESEM, AFM, and TEM techniques. These studies show the formation of globular self-assembled structures in the solid phase of **1b** and the multi-layered sheet-like structure in its anion-bound form. The anion-induced supramolecular polymeric systems were further utilized for ion transport studies across the lipid bilayer membrane. The dose-dependent ion transport activity in HPTS-based vesicular studies furnish the $EC_{50}$ value of 10.9 nM and Hill coefficient $n$ value of 1, indicating a stable supramolecular ion channel formation. The detailed mechanistic ion transport studies confirm the operation of an antiport mechanism via Cl⁻/NO$_3$⁻ exchange across the lipid bilayer. Eventually, the transport of ions through the formation of the ion channel structures was confirmed by performing the planar bilayer conductance measurement experiments. We believe the current work on the stimuli-responsive supramolecular ion transport system with multifunctional properties would provide a way to develop smart materials with efficient bio-medical applications.

## Methods
### General methods
All reagents used for the synthesis were purchased either from Sigma-Aldrich, Avra, TCI, Spectrochem, and used without further purification. Dry solvents, e.g., THF, CHCl₃, CH₂Cl₂, and MeOH used for the synthesis were purchased from Merck and used without further drying. All ¹H and ¹³C NMR spectra were recorded using solution of the compound in deuterated solvents, on either Jeol 400 MHz or Bruker 400 MHz NMR spectrometers. The chemical shifts ($\delta$, in ppm unit) were referenced to the residual signals of deuterium solvents (¹H NMR CDCl₃: $\delta$ 7.26 ppm; ¹³C NMR CDCl₃: $\delta$ 77.2 ppm; ¹H NMR C₂D₂Cl₄: $\delta$ 6.00 ppm; ¹³C NMR C₂D₂Cl₄: $\delta$ 73.8 ppm; ¹H NMR DMSO-$d_6$: $\delta$ 2.5 ppm; ¹³C NMR DMSO-$d_6$: $\delta$ 39.5 ppm). The multiplicities of the peaks are s (singlet), d (doublet), t (triplet), q (quartet), dd (doublet of doublet), m (multiplet). The high-resolution mass spectra (HRMS) were acquired from MicroMass ESI-TOF MS spectrometer and were acquired in the ESI (+ve or −ve) mode. All the fluorescence emission spectra were recorded on a Fluoromax-4 instrument, from Horiba scientific, where experimental cell is equipped with a black injector port and a magnetic stirrer. Planar lipid bilayer conductance measurements were carried out on a workstation from Warner instrument, USA. The single crystal X-ray diffraction (SCXRD) data collected on a Bruker Smart Apex Duo diffractometer using Mo Kα radiation for all the compounds at either 100 K or 150 K temperature. The field emission scanning electron microscopy (FESEM) images data were obtained using FEI Quanta 3D dual beam ESEM at 3.0 kV. The atomic force microscopy (AFM) images were recorded using Nano Wizard Atomic Force Microscopy. The high-resolution transmission electron microscopy (HRTEM) images were acquired on Jeol USA JEM-2200 FS transmission electron microscope.

## Data availability
A detailed description of the findings with complete characterization data can be found in the main text and the Supplementary Information. The X-ray crystal data generated in this study have been deposited in the Cambridge Crystallographic Data Center (CCDC) under accession code 2101675 (**1a**), 2101678 (**1b**), and 2101679 (**1b**_Cl⁻). Source data are provided with this paper.

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

## Acknowledgements

This work was supported by DST, Govt. of India (Under Indo-Russian Collaborative Project Scheme, DST/INT/RUS/RSF/P-24). D.M. and M.A. thank UGC, Govt. of India, for research fellowships. B.D. thanks SERB, Govt. of India for NPDF fellowship (PDF/2020/002670). A.M. thanks Prime Minister's Research Fellowship.

## Author contributions

P.T. conceived and directed the research project. D.M. synthesized and characterized the compounds. D.M. and M.A. have performed the characterization experiment of double-helix and polymeric structure. D.M. and B.D. conducted and analyzed the crystallographic study. D.M. and A.M. performed the ion transport experiments. D.M. and P.T. wrote the paper. All authors approve the final version of the manuscript.

## Competing interests

The authors declare no competing interests.
