## [Peer Review File · Nature Communications]

Formation of supramolecular channels by reversible unwinding-rewinding of bis(indole) double helix via ion coordinationREVIEWER COMMENTS

Reviewer #1 (Remarks to the Author):

This work reports a stimuli responsive bis(indole) derivatives capable of converting between a double helix to a supramolecular polymer. Unwinding of the double helix to polymer can be achieved using chloride salts, conversion back to the double helix is obtained in the presence of silver salts. The double helix was confirmed by solution and solid-state studies, whereas the polymer was confirmed by solid state studies. Lastly, ion transport was studied for these derivatives.

Reviewer #2 (Remarks to the Author):

This manuscript by the group of Talukdar described a series of bis-indoles, which pair into a double helix, but upon addition of chloride the double helix is broken up and supramolecular sheets are formed. In the solid state, it is shown that a channel like structure is formed by the stacked sheets and this motivated the authors to test their compounds in membrane transport, where channel formation facilitates diffusion of anions across the lipid bilayer. Overall, the observed double helix formation as well as the transition to sheet formation upon chloride addition is very interesting. It is, however, nicely confirmed in the solid state by single crystal X-ray analysis while it is not evident that these structures exist in solution. Planar lipid bilayer experiments hint toward channel formation, but what is concluded on the basis of the mechanistic studies with HPTS and lucigenin assays could use additional explanation. Although not the very first example of anion-induced self-assembled channels, as claimed by the authors, it is still a nice study that deserves publication in Nature Communications after the comments below have been properly addressed.

- Page 6: ESI-MS does not give conformational information and does thus not confirm double helix formation. It merely gives an indication of dimer formation in solution. Together with the single NOE contact it is actually the only evidence for double helix formation in solution.

- Page 6: It is highly surprising to see that the three ¹H-NMR experiments performed (dilution, DMSO addition and variable temperature) do not show dissociation of the double helix and therefore a stability constant cannot be given. In the presence of chloride, the dimer breaks up easily. The authors could perform the competitive and VT NMR experiments at higher dilution than 5 mM, use a more competitive additive, or gain insight into the association constant through the use of a ITC dilution experiment.

- Page 10-11 ESI-MS shows the mass of several oligomers, but is no evidence of polymeric structures in solution. DOSY NMR should be used as additional support.

- ESI Figure S20 and Table S6 needs more proper analysis: What is the repeating unit of the supramolecular polymer envisioned? There seems to be three distributions with a regular repeating unit. To what structure does this correspond?

- Page 13: Importantly, it is NOT the first example of anion-induced self-assembled supramolecular channels, see: *Angew. Chem. Int. ed.* 2020, 59, 18920-18926

- Page 13: This will not be the outcome of Lipinski's rule, which has more components than $\log P < 5$.

- Page 14: The hill coefficient n ranges from 0.75 – 0.9 and is thus always lower than 1. This would mean that the channel becomes inactive at higher concentrations, likely through precipitation.

- Page 14: The current vs voltage plot is fitted to a linear (ohmic) relationship. It seems however that it displays a sigmoidal curve. The authors should try to fit the data to different models.

- Page 15: In the presence of valinomycin and KCl as external buffer a K⁺ gradient is created and it

seems the need for NO₃ antiport is ruled out here. Does this mean NO₃ transport is rate-limiting? More explanation is needed on how this confirms the antiport mode of operation.

- In the transport experiments, the compound is added in DMSO solution. How do the authors envision channel formation in the membrane. Will it make a difference if the compound is mixed with chloride first and then added to the vesicle solution?

- ESI Fig. S12 amount of DMSO is incorrectly in mL instead of μ L

- ESI page S28 – the lipid film is vortexed 4-5 times over the course of 1 hour. This should be clarified. How long is the film vortexed and how long is it allowed to rest in between

- Page S29 What is the concentration of lipids in the fluorescence measurement?

- Page S30-31 Y is the fluorescence intensity after addition of excess transporter molecule. This value should correspond to the fluorescent intensity after lysis, but it seems the transport activity at the highest concentration is taken as 1, while this might actually correspond to a concentration of 0.6 such as in the case of 1c.

Reviewer #3 (Remarks to the Author):

The manuscript describes self-assembly, anion-coordination and transmembrane anion transport of a small library of bis(indole) isophthalimides. These molecules form double helix homo- or hetero-dimeric assemblies in solution and solid state. Chloride binding to the central isophthalimide pocket led to unwinding of the double helix and the formation of channel-like anion-bridged supramolecular polymeric stacks. The bis(indole) isophthalimides can transport anions in liposomal and planar lipid bilayer experiments. The development of artificial molecular assemblies that mimic the DNA double helix and its reversible dissociation into single strands is an important goal in the field of supramolecular chemistry. The merits of the current work include the use of easily synthesized small molecules to achieve biomimetic self-assembly, the reversible control of self-assembly using anion-coordination, and the demonstration of biomedically-relevant transmembrane anion transport function using these assemblies. While the current results are interesting, the following issues should be addressed before publication of the manuscript.

1. The formation of anion-bridged supramolecular polymers was demonstrated by solid-state X-ray analysis. However, there was insufficient evidence that such polymers existed in solution. The observation of polymeric species in ESI-MS could correspond to a minor species in the solution with the majority of the anion complexes existing as monomers. Diffusion NMR is a standard technique that would provide unambiguous evidence of supramolecular polymerization in solution.

2. While anion transport was evidenced by the HPTS assay and channel formation supported by the observation of single channel current recording, the bundled ion channel structure in the membrane as inferred from crystal structures and schematically shown in Fig. 4b lacks experimental evidence. In the solid-state and perhaps in CHCl₃ or CH₂Cl₂ solutions, supramolecular polymerization leading to channel-like assemblies was driven by Cl⁻ binding which linked adjacent isophthalimides into sheets. However, in lipid bilayer membranes in water, Cl⁻ binding is extremely weak as Cl⁻ is strongly hydrated in water. Therefore, it is unlikely that the same structures observed in crystals would form in lipid bilayers. Single column channels, or other unidentified assembly modes could be responsible for the anion conductance observed. I appreciate the difficulty of studying molecular self-assembly in the membrane - if the authors could not provide direct evidence of the structure of ion channels in the membrane, it would be prudent to re-draw Fig.4, perhaps only representing single-column channels. The statement of “anion-induced self-assembled supramolecular ion channel formation across the lipid bilayer membrane” should be removed as the channel formation is likely anion-independent due to anion binding being extremely weak in water.

3. As discussed theoretically by Berezin (Supramol. Chem. 2013, 25, 323) and demonstrated experimentally by Gale (Chem. Commun. 2021, 57, 3979), the apparent selectivity determined by varying the extraventricular anion in HPTS base pulse assay is opposite to the true selectivity. To

address the anion selectivity, the authors could employ modified methods by Gale or use an acid-pulse instead of a base pulse (see Chem. Commun. 10.1039/D2CC00144F). Since anion selectivity is not the focus of the manuscript, without the additional experiments, the term "selectivity order" (line 296) can be replaced by "pH dissipation trends in the order". In addition, anions like acetate and fluoride should not be used in HPTS assays because the conjugate acids of these anions are membrane permeable and exist in significant amounts at pH 7. The HOAc simple diffusion from outside to the inside of liposomes opposed the direction of H⁺ efflux along the pH gradient, leading to the weakest fluorescence change when OAc⁻ was used as the extravesicular anion (Fig. 5c). This originates from an acid diffusion process and does not provide any information of OAc⁻ selectivity.

4. Page S37, "planner" should be "planar"

Reply to Reviewers' Comment

Comments to the Author

Reviewer #1 (Remarks to the Author):

This work reports a stimuli responsive bis(indole) derivatives capable of converting between a double helix to a supramolecular polymer. Unwinding of the double helix to polymer can be achieved using chloride salts, conversion back to the double helix is obtained in the presence of silver salts. The double helix was confirmed by solution and solid-state studies, whereas the polymer was confirmed by solid state studies. Lastly, ion transport was studied for these derivatives.

Response: We are highly thankful to the reviewer for giving his/her precious time and carefully reading our work for the evaluation of the manuscript. We have also received some constructive comments from other reviewers, where they have suggested to perform some experimental studies. Based on their comments, we have carried out necessary experiments, and the outcome of the investigations are addressed below (Also included in the revised manuscript and supplementary information files).

Reviewer #2 (Remarks to the Author):

This manuscript by the group of Talukdar described a series of bis-indoles, which pair into a double helix, but upon addition of chloride the double helix is broken up and supramolecular sheets are formed. In the solid state, it is shown that a channel like structure is formed by the stacked sheets and this motivated the authors to test their compounds in membrane transport, where channel formation facilitates diffusion of anions across the lipid bilayer. Overall, the observed double helix formation as well as the transition to sheet formation upon chloride addition is very interesting. It is, however, nicely confirmed in the solid state by single crystal X-ray analysis while it is not evident that these structures exist in solution. Planar lipid bilayer experiments hint toward channel formation, but what is concluded on the basis of the mechanistic studies with HPTS and lucigenin assays could use additional explanation. Although not the very first example of anion-induced self-assembled channels, as claimed by the authors, it is still a nice study that deserves publication in Nature Communications after the comments below have been properly addressed.

Response: We are highly thankful to the reviewer for carefully reading our work for the evaluation of the manuscript. We also thank the reviewer for appreciating our work. Based on the comments of the reviewer, we have gone through extensive literature survey, carried out necessary experiments, and the outcome of the investigations are addressed below. We have also prepared a revised version of the manuscript and supplementary files after incorporating the suggestions/comments received from the other reviewers. In the section below, we are providing a point-by-point reply to all comments.

Question 1: Page 6: ESI-MS does not give conformational information and does thus not confirm double helix formation. It merely gives an indication of dimer formation in solution. Together with the single NOE contact it is actually the only evidence for double helix formation in solution.

Response: We thank the reviewer for the comment. For the characterization of the double helix structure in the original version of the manuscript, formed by the bis(indole) molecules in solution phase, we have proceeded in a stepwise manner. Initially, we have performed the ESI-MS studies of **1a**, **1b**, and (**1a+1b**) compounds, where in all the instances dimeric structure formation has been observed (Figure S4, S5, and S6) and that has been claimed in the previous submission. It is to be noted that the ESI-MS analysis data has been used by Prof. Huc and co-workers for the characterization of double helix structure ((a) Gan, Q., Li, F., Li, G., Kauffmann, B., Xiang, J., Huc, I. & Jiang, H. *Chem. Commun.* **46**, 297-299 (2010); (b) Shang, J., Gan, Q., Dawson, S. J., Rosu, F., Jiang, H., Ferrand, Y. & Huc, I. *Org. Lett.* **16**, 4992-4995 (2014)). The claim of double helix structure from the ESI-MS study is supported by the fact that the most feasible way to form a dimeric aggregate is the formation of double helix structure, where all possible non-covalent interactions can be utilized for the stabilisation of the dimeric complex. However, additional supportive studies have also been performed.

Due to the σ -symmetric structure of our double helix-forming molecules, we could not conclude double helix structure formation from the ^1H NMR and NOESY NMR solution phase studies for single molecule. Therefore, we have performed these studies (Figure S9 for ^1H NMR and Figure S10 for NOESY NMR) for the mixture of molecules i.e. (**1c** + **1d**), which evidenced the formation of double helix structure in solution phase.

Fig. S14. The inter-double helical hydrogen bonding interactions in the crystal structure of **1b** (A) and in the cartoon representation (B), through $\text{N-H}\cdots\text{O}=\text{C}$ bonding. The ^1H NMR spectra of double-helix forming molecule **1c** in absence and presence of urea and cyanuric acid (C).

It is well known from the literature that the addition of hydrogen bond forming molecules such as urea leads to the decrease in melting point temperature of the DNA double helix structure ((a) Nordstrom, L. J., Clark, C. A., Andersen, B., Champlin, S. M. & Schweinfus, J. J. *Biochemistry* **45**, 9604-9614 (2006); (b) Oprzeska-Zingrebe, E. A. & Smiatek, J. *Biophys. J.* **114**, 1551-1562 (2018)). To observe the effect of urea in the double helix structure formation, ^1H NMR experiments have been performed for the compound **1c** in the presence and absence of urea, where the addition of urea leads to chemical shift change of only the H_i proton without affecting other protons (Fig. S14B). This observation is supported by the fact that in the absence of urea the bis(indole) molecules form double helix structure employing the intermolecular hydrogen bonding interaction, where the H_i proton is not involved in these interactions. And therefore the H_i proton participates in the inter-double helical bonding interaction through $\text{N}-\text{H}_i\cdots\text{O}=\text{C}$ bonding (Fig. S14A). With the addition of urea this $\text{N}-\text{H}_i\cdots\text{O}=\text{C}$ bonding interaction got disturbed and leads to the downfield shift of the only H_i proton without affecting other acidic protons. These combined observations also support that our bis(indole) molecule forms double helix structure in solution, otherwise all acidic proton should have been either downfield or upfield shifted.

Question 2: Page 6: It is highly surprising to see that the three ^1H -NMR experiments performed (dilution, DMSO addition and variable temperature) do not show dissociation of the double helix and therefore a stability constant cannot be given. In the presence of chloride, the dimer breaks up easily. The authors could perform the competitive and VT NMR experiments at higher dilution than 5 mM, use a more competitive additive, or gain insight into the association constant through the use of a ITC dilution experiment.

Response: We are extremely thankful to the reviewer for the suggestion. In the original version of the manuscript, we showed the stability of the double helix structure upon dilution, DMSO- d_6 addition, and in variable temperature experiments. However, in all these instances preservation of the double helix structure has been observed, where the experiments were performed at higher concentration, i.e. 5 mM receptor concentration. Based on the suggestion

Fig. 1g. The partially stacked ^1H NMR spectra of **1c** (0.1 mM) in $\text{C}_2\text{D}_2\text{Cl}_4$ at variable temperature (263K–373K).

of the reviewer, we have performed the similar variable temperature experiment (263K–373K) at a lower concentration of receptor **1c** (0.1 mM, 50 times diluted compared to the previous experiment). Upon increasing the temperature of the diluted solution of the double helix forming molecule, the upfield shift of NH protons (chemical shift) has been observed (Fig. 1g), which indicates the dissociation of the double helix structure to its monomeric form (Misra, R., Dey, S., Reja, R. M. & Gopi, H. N. *Angew. Chem., Int. Ed.* **57**, 1057-1061 (2018)).

We have also performed the ^1H NMR experiments with the addition of more competitive additives such as urea and cyanuric acid (forms hydrogen bonds through $\text{N}-\text{H}\cdots\text{O}=\text{C}$ interactions), based on the suggestion of the reviewer. However, in both the experiments we have not observed any dissociation of double helix structure, rather the inter-helical non-covalent bond getting disturbed, which is reflected in the downfield shift of H_i proton (Fig. S14B).

Fig.S14B. The ^1H NMR spectra of double-helix forming molecule **1c** and **1c** with urea and cyanuric acid.

As we could not calculate the binding constant value from the ^1H NMR titration experiment, based on the suggestion of the reviewer, we have performed the ITC titration experiment to get some idea about association constant value with the addition of tetrabutylammonium chloride (TBACl). For the ITC experiment, a 100 μM solution of the receptor (280 μL) in methanol was titrated with 1 mM of TBACl solution in methanol.

Fig. 1. The ITC titration experiment of **1c** in solution (100 μM) of methanol, with tetrabutylammonium chloride (TBACl) solution (1 mM) of the same solvent mixture at 298 K.

From the observed titration plot data, we could not get any rational association constant value due to use of methanol for the preparation of stock and titrating solution. Moreover, due to the limitation of the instrument (MicroCal PEAQ-ITC from Malvern Panalytical) we could not perform the titration experiment using other polar-aprotic solvent. However, to calculate the association constant value, UV-Vis titration experiment has been conducted in $C_2H_2Cl_4$ solvent system. The addition of TBACl salt to bis(indole) **1c** solution ($10\ \mu\text{M}$) provided a significant intensity decrement of a shoulder peak at 352 nm of the UV-Vis absorption spectra (Fig. S27), indicating the anionic complex formation by the receptor with the addition of Cl^- . The association constant was calculated by fitting the UV-Vis absorption data in the BindFit model (supramolecular.org) and is found to be $1.26 \times 10^5\ \text{M}^{-1}$ ($\pm 10.7\%$).

Fig. S27. The UV-Vis titration experiment of bis(indole) compound **1c** with the addition of TBACl in $C_2H_2Cl_4$ at 25 C (left side). Screenshot of the fitted data plot from supramolecular.org: The binding constant was found to be 1.26×10^5 ($\pm 10.7\%$) M^{-1} in 1:1 receptor to anion binding model. The changing pattern of chemical shift and chemical shift residuals with the increasing equivalent TBACl (right side). The Bindfit URL for this experiment is: <http://app.supramolecular.org/bindfit/view/42f6762e-221a-4e81-bbf6-994aefd74cca>.

The above-mentioned experimental observations and discussion have been incorporated in the revised version of the manuscript and supplementary information files.

Question 3: Page 10-11 ESI-MS shows the mass of several oligomers, but is no evidence of polymeric structures in solution. DOSY NMR should be used as additional support.

Response: We thank the reviewer for the suggestion. Based on the suggestion of the reviewer to check the aggregation behaviour, we have gone through some literature reports ((a) Zhan, T.-G., Zhou, T.-Y., Qi, Q.-Y., Wu, J., Li, G.-Y. & Zhao, X. *Polym. Chem.* **6**, 7586-7593 (2015); (b) Budak, A. & Aydogan, A. *Chem. Commun.* **57**, 4186-4189 (2021)), where they have performed the 2D-DOSY NMR experiments for the characterization of anion induced supramolecular polymer in solution phase. In their experiments, they have observed a decrease in diffusion coefficient value with the increasing concentration of the receptor, which gives evidence in support of supramolecular polymer formation in the solution phase. Following the above-mentioned literature reports, we have also performed the 2D-DOSY NMR experiment for the free and anionic complex of the receptor CHCl_3 solution. At first 2D-DOSY NMR spectrum was recorded for the 5.0 mM solution of the free bis(indole)

compound **1c**, which provided the diffusion coefficient value of $5.96 (\pm 0.03) \times 10^{-6} \text{ cm}^2/\text{s}$ (Fig. S28). With the addition of 3.0 equivalent TBACl salt in the same 5.0 mM receptor solution leads the decrement of diffusion coefficient value, which is found to be $4.25 (\pm 0.02) \times 10^{-6} \text{ cm}^2/\text{s}$ (Fig. S29). This decrease in the diffusion coefficient value supports the polymer structure formation in the solution phase, which is formed by the conversion of the double helix form to monomeric unit and subsequent formation of polymeric aggregates.

Further, the 2D-DOSY NMR spectra were recorded at higher concentrations of receptor with 3.0 equivalent of TBACl. The diffusion coefficient values have been calculated and the values are $3.14 (\pm 0.02) \times 10^{-6} \text{ cm}^2/\text{s}$ and $2.64 (\pm 0.02) \times 10^{-6} \text{ cm}^2/\text{s}$ for 14 mM and 20 mM concentration of receptor (with 3.0 equivalent of TBACl), respectively (Fig. S30 and S31). Overall, in the presence of Cl^- ion, the diffusion coefficient values decreased from $4.25 (\pm 0.02) \times 10^{-6}$ to $2.64 (\pm 0.02) \times 10^{-6} \text{ cm}^2/\text{s}$ with the increasing concentration of bis(indole) compound **1c**, from 5 to 20 mM.

The above-mentioned observations and discussions have been incorporated into the revised version of the manuscript and supplementary information files.

Fig. S28. The 2D DOSY NMR spectrum of bis(indole) compound **1c** (5 mM) in CHCl_3 at 25 °C. Two sub-figures in the bottom sections are the screenshots of the processed data. The diffusion coefficient value for this compound in the above-mentioned condition is $5.96 (\pm 0.03) \times 10^{-6} \text{ cm}^2/\text{s}$.

Fig. S29. The 2D DOSY NMR spectrum of bis(indole) compound **1c** (5 mM) with 3 equiv. of TBACl in CHCl_3 at 25 °C. Two sub-figures in the bottom sections are the screenshots of the processed data. The diffusion coefficient value for this compound in the above-mentioned condition is $4.25 (\pm 0.02) \times 10^{-6} \text{ cm}^2/\text{s}$.

Fig. S30. The 2D DOSY NMR spectrum of bis(indole) compound **1c** (14 mM) with 3 equiv. of TBACl in CHCl_3 at 25 °C. Two sub-figures in the bottom sections are the screenshots of the processed data. The diffusion coefficient value for this compound in the above-mentioned condition is $3.14 (\pm 0.02) \times 10^{-6} \text{ cm}^2/\text{s}$.

Fig. S31. The 2D DOSY NMR spectrum of bis(indole) compound **1c** (20 mM) with 3 equiv. of TBACl in CHCl₃ at 25 °C. Two sub-figures in the bottom sections are the screenshots of the processed data. The diffusion coefficient value for this compound in the above-mentioned condition is $2.64 (\pm 0.02) \times 10^{-6} \text{ cm}^2/\text{s}$.

Question 4: ESI Figure S20 and Table S6 needs more proper analysis: What is the repeating unit of the supramolecular polymer envisioned? There seems to be three distributions with a regular repeating unit. To what structure does this correspond?

Response: We thank the reviewer for the suggestion. Based on the advice of the reviewer, we have re-analyzed the ESI-MS data. We also apologize for our mistake for the wrong interpretation of the polymeric peak analysis. The ESI-MS spectrum was recorded for compound **1b** with 3.0 equivalents of tetramethylammonium chloride (TMACl) in positive (+ve) mode. While analyzing the mass spectrum data, the calculated masses (Table S6) are observed as major signals in the recorded spectrum (Fig. S21), where the computed pattern is found to follow a nice repeating trend. In the repeating trend, the compounds are bound with Cl⁻ and the TMA⁺ ion to make it either singly (+1) or doubly (+2) positively charged species. As we have recorded the spectrum with the m/z value of up to 2000, the singly charged species of higher complex has not been observed.

The answer to the question “What is the repeating unit of the supramolecular polymer envisioned?”: The repeating unit is TMACl. For example, the peak at m/z = 843.41 corresponds to [M + Cl + 2TMA]⁺ and that at 952.47 corresponds to [M + 2Cl + 3TMA]⁺. So the difference is TMACl = 109.06. However, one needs to see the presence of different oligomeric peaks, which are evident from the ESI-MS data.

While trying to match the m/z distribution patterns with the probable aggregated structures, it was observed that at lower m/z values, the doubly positive charged species appeared. On moving towards a higher m/z value, the singly charged followed by doubly

charged peaks were observed, where the repeating unit of TMACl separates the peaks. On the other hand, at a higher m/z value (starting from $m/z = 1007.01$), different sets of aggregated doubly charged species were observed. It should be noted that even though we marked the peak maxima (in Fig. S21) with single aggregated oligomeric structures, every single peak could correspond to a different combination of aggregated units. For example, the peak at $m/z = 731.41$ corresponds to $[M + 6Cl + 8TMA]^{2+}$ and the peak at $m/z = 732.39$ corresponds to $[2M + 6Cl + 9TMA]^{3+}$ appeared at very close region on the spectrometry data. Even though we could not analyse all the small signals from the spectrometry data, looking at the different combinations of oligomeric distribution patterns confirmed the formation of supramolecular polymer with TMACl.

Fig. S21. The ESI-MS spectrum of **1b** with TMACl in positive (+ve) mode.

Table S6. Calculated and observed ESI-MS mass of **1b** with Cl^- (where $M = \mathbf{1b}$).

Sr. No.	General Formula (For M, 2M, 3M, 4M and 5M)	m/z	Sr. No.
1	$[M + n Cl + (n+1)TMA]^+$	$[M + Cl + 2TMA]^+$	843.41
	(where $n = 1, 2, 3, 4$ or 5)	$[M + 2Cl + 3TMA]^+$	952.47

		$[M + 3Cl + 4TMA]^+$	1061.54
		$[M + 4Cl + 5TMA]^+$	1170.61
		$[M + 5Cl + 6TMA]^+$	1279.67
2	$[M + n Cl + (n+2)TMA]^{2+}$ (where $n = 2, 3, 4, 5$ or 6)	$[M + 2Cl + 4TMA]^{2+}$	513.28
		$[M + 3Cl + 5TMA]^{2+}$	567.81 (Low)
		$[M + 4Cl + 6TMA]^{2+}$	622.35
		$[M + 5Cl + 7TMA]^{2+}$	676.88
		$[M + 6Cl + 8TMA]^{2+}$	731.41
3	$[2M + n Cl + (n+1)TMA]^+$ (where $n = 1, 2, 3, 4$ or 5)	$[2M + Cl + 2TMA]^+$	1503.66
		$[2M + 2Cl + 3TMA]^+$	1612.73
		$[2M + 3Cl + 4TMA]^+$	1721.79
		$[2M + 4Cl + 5TMA]^+$	1830.86
		$[2M + 5Cl + 6TMA]^+$	1939.92
4	$[2M + n Cl + (n+2)TMA]^{2+}$ (where $n = 2, 3, 4, 5, 6, 7$ or 8)	$[2M + 2Cl + 4TMA]^{2+}$	843.41
		$[2M + 3Cl + 5TMA]^{2+}$	897.94
		$[2M + 4Cl + 6TMA]^{2+}$	952.47
		$[2M + 5Cl + 7TMA]^{2+}$	1007.01
		$[2M + 6Cl + 8TMA]^{2+}$	1061.54
		$[2M + 7Cl + 9TMA]^{2+}$	1116.08
		$[2M + 8Cl + 10TMA]^{2+}$	1170.61
5	$[3M + n Cl + (n+1)TMA]^+$ (where $n = 2, 3,$ or higher)	Above 2000	Above 2000
6	$[3M + n Cl + (n+2)TMA]^{2+}$ (where $n = 2, 3, 4, 5, 6, 7, 8, 9,$	$[3M + 2Cl + 4TMA]^{2+}$	1173.53
		$[3M + 3Cl + 5TMA]^{2+}$	1228.07

	10 or 11)	$[3M + 4Cl + 6TMA]^{2+}$	1282.60
		$[3M + 5Cl + 7TMA]^{2+}$	1337.13
		$[3M + 6Cl + 8TMA]^{2+}$	1391.67
		$[3M + 7Cl + 9TMA]^{2+}$	1446.20
		$[3M + 8Cl + 10TMA]^{2+}$	1500.73
		$[3M + 9Cl + 11TMA]^{2+}$	1555.27
		$[3M + 10Cl + 12TMA]^{2+}$	1609.80
		$[3M + 11Cl + 13TMA]^{2+}$	1664.33
7	$[4M + n Cl + (n+1)TMA]^+$ (where $n = 3, 4,$ or higher)	Above 2000	Above 2000
8	$[4M + n Cl + (n+2)TMA]^{2+}$ (where $n = 3, 4, 5, 6, 7, 8, 9, 10$ or 11)	$[4M + 3Cl + 5TMA]^{2+}$	1558.19
		$[4M + 4Cl + 6TMA]^{2+}$	1612.73
		$[4M + 5Cl + 7TMA]^{2+}$	1667.26
		$[4M + 6Cl + 8TMA]^{2+}$	1721.79
		$[4M + 7Cl + 9TMA]^{2+}$	1776.32
		$[4M + 8Cl + 10TMA]^{2+}$	1830.86
		$[4M + 9Cl + 11TMA]^{2+}$	1885.39
		$[4M + 10Cl + 12TMA]^{2+}$	1939.92
		$[4M + 11Cl + 13TMA]^{2+}$	1994.46
9	$[5M + n Cl + (n+1)TMA]^+$ (where $n = 4, 5,$ or higher)	Above 2000	Above 2000
10	$[5M + n Cl + (n+2)TMA]^{2+}$ (where $n = 4, 5$ or 6)	$[5M + 4Cl + 6TMA]^{2+}$	1942.85
		$[5M + 5Cl + 7TMA]^{2+}$	1997.38
		$[5M + 6Cl + 8TMA]^{2+}$	Above 2000

This time we have also recorded the ESI-MS data for the same combination of compound (i.e. **1b** with TMACl salt) in negative (–ve) mode. The calculated masses (Table S7) are observed as major signals in the recorded spectrum (Fig. S22), where the computed pattern is found to follow a nice repeating trend. In the repeating trend, the compounds are bound with Cl[–] and the TMA⁺ ion to make it singly (–1) or doubly (–2) negatively charged species. The peaks corresponding to different aggregated polymeric units were observed from the ESI-MS experiment and are listed therein below table (Table S7; where, M is the exact mass of **1b**).

Fig. S22: The ESI-MS spectrum of **1b** with Cl[–] in negative (–ve) mode.

Table S7. Calculated and observed ESI-MS mass of **1b** with Cl[–] (where M = **1b**) in negative (–ve) mode.

Sr. No.	General Formulae (For M, 2M, 3M, 4M and 5M)	m/z (–ve mode)	Calculated & Matched Signals
1	[M + n TMA + (n+1)Cl] [–] (where n = 1, 2, 3 or 4)	[M + TMA + 2Cl] [–]	804.28
		[M + 2TMA + 3Cl] [–]	913.35
		[M + 3TMA + 4Cl] [–]	1022.42

		$[M + 4TMA + 5Cl]^-$	1131.48
2	$[M + n TMA + (n+2)Cl]^{2-}$ (where $n = 0, 1$ or 2)	$[M + 2Cl]^{2-}$	365.09
		$[M + TMA + 3Cl]^{2-}$	419.62
		$[M + 2TMA + 4Cl]^{2-}$	474.15
3	$[2M + n TMA + (n+1)Cl]^-$ (where $n = 1, 2, 3, 4$ or 5)	$[2M + TMA + 2Cl]^-$	1464.53
		$[2M + 2TMA + 3Cl]^-$	1573.60
		$[2M + 3TMA + 4Cl]^-$	1682.66
		$[2M + 4TMA + 5Cl]^-$	1791.73
		$[2M + 5TMA + 6Cl]^-$	1900.80
4	$[2M + n TMA + (n+2)Cl]^{2-}$ (where $n = 1, 2, 3, 4, 5$ or 6)	$[2M + TMA + 3Cl]^{2-}$	749.75
		$[2M + 2TMA + 4Cl]^{2-}$	804.28
		$[2M + 3TMA + 5Cl]^{2-}$	858.81
		$[2M + 4TMA + 6Cl]^{2-}$	913.35
		$[2M + 5TMA + 7Cl]^{2-}$	967.88
		$[2M + 6TMA + 8Cl]^{2-}$	1022.42
5	$[3M + n TMA + (n+1)Cl]^-$ (where $n = 2, 3$, or higher)	Above 2000	Above 2000
6	$[3M + n TMA + (n+2)Cl]^{2-}$ (where $n = 0, 1, 2, 3, 4, 5, 6$ or 7)	$[3M + 2Cl]^{2-}$	1025.34
		$[3M + TMA + 3Cl]^{2-}$	1079.88
		$[3M + 2TMA + 4Cl]^{2-}$	1134.41
		$[3M + 3TMA + 5Cl]^{2-}$	1188.94
		$[3M + 4TMA + 6Cl]^{2-}$	1243.47
		$[3M + 5TMA + 7Cl]^{2-}$	1298.01
		$[3M + 6TMA + 8Cl]^{2-}$	1352.54

		$[3M + 7TMA + 9Cl]^{2-}$	1407.07
7	$[4M + n TMA + (n+1)Cl]^-$ (where $n = 3, 4$ or higher)	Above 2000	Above 2000
8	$[4M + n TMA + (n+2)Cl]^{2-}$ (where $n = 1, 2, 3, 4, 5, 6, 7, 8$ or 9)	$[4M + TMA + 3Cl]^{2-}$	1410.00
		$[4M + 2TMA + 4Cl]^{2-}$	1464.53
		$[4M + 3TMA + 5Cl]^{2-}$	1519.07
		$[4M + 4TMA + 6Cl]^{2-}$	1573.60
		$[4M + 5TMA + 7Cl]^{2-}$	1628.13
		$[4M + 6TMA + 8Cl]^{2-}$	1682.66
		$[4M + 7TMA + 9Cl]^{2-}$	1737.20
		$[4M + 8TMA + 10Cl]^{2-}$	1791.73
	$[4M + 9TMA + 11Cl]^{2-}$	1846.26	
9	$[5M + n TMA + (n+1)Cl]^-$ (where $n = 4, 5$ or higher)	Above 2000	Above 2000
10	$[5M + n TMA + (n+2)Cl]^{2-}$ (where $n = 2, 3, 4,$ or 5)	$[5M + 2TMA + 4Cl]^{2-}$	1794.65
		$[5M + 3TMA + 5Cl]^{2-}$	1849.18
		$[5M + 4TMA + 6Cl]^{2-}$	1903.72
		$[5M + 5TMA + 7Cl]^{2-}$	1958.25

Question 5: Page 13: Importantly, it is NOT the first example of anion-induced self-assembled supramolecular channels, see: *Angew. Chem. Int. ed.* 2020, 59, 18920-18926.

Response: We thank the reviewer for pointing out our mistake. Based on the suggestion of the reviewer, we have modified the introduction part where “the first report of anion-induced self-assembled supramolecular channel” has been claimed. Moreover, we have cited the suggested reference in the supramolecular channel discussion part.

Question 6: Page 13: This will not be the outcome of Lipinski’s rule, which has more components than $\log P < 5$.

Response: We thank the reviewer for the comment. We agree with the reviewer that the Lipinski's rule is applicable for the molecule with logP value less than 5. However, in our cases, the transport activity got decreased with the increasing logP values, which may be due to the increased hydrophobicity of the transporter molecules. This phenomenon we have written in a way like more the deviation of the logP value from 5, less the ion transport activity. However, this phenomenon has a very little link with the Lipinski's rule. Therefore, based on the suggestion of the reviewer and considering the above-mentioned discussion, we have removed the "Lipinski's rule" term from the discussion part and the revised phrases are mentioned below.

"The above ion transport activity sequence of **1b** > **1a** > **1c** > **1d** may be the outcome of the increasing hydrophobicity of the transporter molecules, which deviates the logP values of the compound far away from 5. The octyl chain bearing compound **1d** with logP value of 11.08 was found to be least active among the four molecules."

Question 7: Page 14: The hill coefficient n ranges from 0.75 – 0.9 and is thus always lower than 1. This would mean that the channel becomes inactive at higher concentrations, likely through precipitation.

Response: We thank the reviewer for the comment. Based on the suggestion of the reviewer, we have gone through the extensive literature survey and found that the Hill coefficient value close to 1 indicates the formation of a stable channel structure (Litvinchuk, S., Bollot, G., Mareda, J., Som, A., Ronan, D., Shah, M. R., Perrottet, P., Sakai, N. & Matile, S. *J. Am. Chem. Soc.* **126**, 10067-10075 (2004)). However, in our system, the Hill coefficient values are ≤ 1 . As per the suggestion of the reviewer, at higher concentrations, the channel-forming molecules are likely to precipitate. However, while performing the ion transport experiments, we have not observed any precipitation in any of the transport experiments.

Question 8: Page 14: The current vs voltage plot is fitted to a linear (ohmic) relationship. It seems however that it displays a sigmoidal curve. The authors should try to fit the data to different models.

Response: We thank the reviewer for the suggestion. In the original version of the manuscript, the current trace values against the potential have been fitted in a linear equation, where we have observed some deviation from the linear equation. To get some idea about the proper fitting of the current traces data, we have gone through the literature report on "synthetic chloride channel" by Prof. Barboiu and co-workers (Zheng, S.-P., Jiang, J.-J., van der Lee, A. & Barboiu, M. *Angew. Chem., Int. Ed.* **59**, 18920-18926 (2020)), where they fit the current traces in exponential equation. Based on the suggestion of the reviewer, we have fit the data in a sigmoidal equation. Now we have incorporated this analysis in the revised version of the manuscript.

Fig. S45. The plot current traces vs voltage obtained from the channel opening data at different potentials of **1b** fit in sigmoidal equation.

Question 9: Page 15: In the presence of valinomycin and KCl as external buffer a K^+ gradient is created and it seems the need for NO_3^- antiport is ruled out here. Does this mean NO_3^- transport is rate-limiting? More explanation is needed on how this confirms the antiport mode of operation.

Response: We thank the reviewer for the comment. In antiport assay, in the presence of valinomycin and KCl, the Cl^- concentration gradient dissipation can occur in various possible ways such as (a) $\text{Cl}^-/\text{NO}_3^-$ antiport, (b) K^+/Cl^- symport by cooperative effect of transporter and valinomycin. However, the cooperative transport of K^+/NO_3^- symport will have comparably less effect, as the concentration of NO_3^- ion is the same in both the intra and extravesicular solution. The synergistic effect of valinomycin and the transporter enhanced the overall transport activity of the transporter in the presence of valinomycin, compared to the transporter alone.

On the other hand, in the symport mechanism, a similar cooperative effect between transporter and valinomycin will not occur as the transporter would be self-sufficient in transporting K^+ along with Cl^- and thereby maintaining the electroneutrality. Therefore, the activity of the transporter molecule will be similar in the presence of valinomycin, compared to the transporter alone.

The above-mentioned explanations confirmed that $\text{Cl}^-/\text{NO}_3^-$ antiport is the primary mode of transport, as we have observed enhanced transport activity in the presence of valinomycin.

Fig. S43. Schematic representation of vesicles for fluorescence kinetics assay showing antiport and symport mechanism in presence of Valinomycin across EYPC-LUVs \Rightarrow Lucigenin.

Question 10: In the transport experiments, the compound is added in DMSO solution. How do the authors envision channel formation in the membrane. Will it make a difference if the compound is mixed with chloride first and then added to the vesicle solution?

Response: We thank the reviewer for the comment. In the HPTS-based ion transport experiment, the transporter molecules were added to the vesicular solution containing 100 mM of NaCl solution and it was expected that the similar channel formation (like the crystal structure) will occur inside the hydrophobic lipid bilayer membrane with the help of Cl⁻ ions from the solution that eventually lead the transport of Cl⁻ ions across the bilayer membrane. Additionally, we also performed the ion transport experiment where the transporter molecule was mixed with the NaCl salt prior to its addition into the vesicular solution. The addition of this chloride bound transporter molecule does not make any significant difference in the ion transport activity as compared to the addition of the free transporter molecule. These observations suggested that the channel formation mode has not changed, when the Cl⁻ ion incorporated transporter stock solution was added for the ion transport activity measurement experiments.

Fig. S37. The ion transport activity in HPTS assay for free and Cl⁻ ion incorporated stock solution of 5.0 nM of bis(indole) compound **1b**.

The above-mentioned experimental observation has been incorporated into the revised version of the manuscript and supplementary information files.

Question 11: ESI Fig. S12 amount of DMSO is incorrectly in mL instead of μL .

Response: We thank the reviewer for pointing out the typo. Now we have replaced the “mL” term with the “ μL ” in the revised version of the supplementary file.

Question 12: ESI page S28 – the lipid film is vortexed 4-5 times over the course of 1 hour. This should be clarified. How long is the film vortexed and how long is it allowed to rest in between.

Response: We thank the reviewer for suggesting to write the clarified words regarding the vortex process of vesicles preparation. The lipid film was vortexed for ~ 2 minutes with 10 minutes of rest in between. And this cycle was repeated for 5 times, where the overall process takes around 1 hour. Based on the suggestion of the reviewer and considering the broader range of readers of this work, we have incorporated the following phrases in the experimental details of the revised supplementary files:

“The thin lipid film was vortexed for 5 times (2 minutes each time) with 10 minutes of rest in between, where the overall process takes around 1 hour.”

Question 13: Page S29 What is the concentration of lipids in the fluorescence measurement?

Response: We thank the reviewer for the comment. The concentration of the lipid in the cuvette during the fluorescence measurement experiment is ~ 65 μM . The above-mentioned concentration value is supported by the following explanation:

“For the preparation of the vesicles, 25 mg of EYPC lipid (Egg PC, Formula Weight = 770.123) has been used, which resulted in 6 mL of vesicles stock solution/suspension of concentration ~ 5.4 mM, considering no significant loss of lipid during the vesicles preparation process. The fluorescence experiments were performed by suspending 25 μL of such stock solution in the 2 mL of buffer solution, which finally furnished the lipid concentration of ~ 65 μM .”

The above-mentioned detail has been incorporated in the revised version of the supplementary information file.

Question 14: Page S30-31 Y is the fluorescence intensity after addition of excess transporter molecule. This value should correspond to the fluorescent intensity after lysis, but it seems the transport activity at the highest concentration is taken as 1, while this might actually correspond to a concentration of 0.6 such as in the case of **1c**.

Response: We thank the reviewer for the suggestion. For the Hill analysis, at first, the maximum intensity value (for a particular compound) was normalized to 1 and then fitted into the Hill equation, which resulted the EC_{50} and Hill coefficient values. However, based on the suggestion of the reviewer, we have re-calculated the Hill analysis data considering the Y as the fluorescent intensity after lysis. The re-calculated Hill analysis plot along with the EC_{50} and Hill coefficient values have been provided below and also incorporated in the revised version of the manuscript and supplementary information files.

Compound	EC_{50} value	Hill coefficient (n)
1a	13.7 nM	0.76
1b	10.9 nM	0.86
1c	341.5 nM	0.90

Fig. S33, S34B, and S35B. The Hill plot analysis for transporter **1b** (A), **1a** (B), and **1c** (C) to get EC_{50} and the Hill coefficient value.

Reviewer #3 (Remarks to the Author):

The manuscript describes self-assembly, anion-coordination and transmembrane anion transport of a small library of bis(indole) isophthalimides. These molecules form double helix homo- or hetero-dimeric assemblies in solution and solid state. Chloride binding to the central isophthalimide pocket led to unwinding of the double helix and the formation of channel-like anion-bridged supramolecular polymeric stacks. The bis(indole) isophthalimides can transport anions in liposomal and planar lipid bilayer experiments. The development of artificial molecular assemblies that mimic the DNA double helix and its reversible dissociation into single strands is an important goal in the field of supramolecular chemistry. The merits of the current work include the use of easily synthesized small molecules to achieve biomimetic self-assembly, the reversible control of self-assembly using

anion-coordination, and the demonstration of biomedically-relevant transmembrane anion transport function using these assemblies. While the current results are interesting, the following issues should be addressed before publication of the manuscript.

Response: We are greatly thankful to the reviewer for careful evaluation of the manuscript and for appreciating our work. Based on the comments of the reviewer, we have gone through some literature reports, carried out necessary experiments which support our model, and the outcome of the investigations are addressed below. We have also prepared a revised version of the manuscript and supplementary files after incorporating the suggestions/comments received from the reviewers. In the section below, we are providing a point-by-point reply to all comments.

Question 1: The formation of anion-bridged supramolecular polymers was demonstrated by solid-state X-ray analysis. However, there was insufficient evidence that such polymers existed in solution. The observation of polymeric species in ESI-MS could correspond to a minor species in the solution with the majority of the anion complexes existing as monomers. Diffusion NMR is a standard technique that would provide unambiguous evidence of supramolecular polymerization in solution.

Response: We thank the reviewer for the suggestion. Based on the suggestion of the reviewer to check the aggregation behaviour, we have gone through some literature reports ((a) Zhan, T.-G., Zhou, T.-Y., Qi, Q.-Y., Wu, J., Li, G.-Y. & Zhao, X. *Polym. Chem.* **6**, 7586-7593 (2015); (b) Budak, A. & Aydogan, A. *Chem. Commun.* **57**, 4186-4189 (2021)), where they have performed the 2D-DOSY NMR experiments for the characterization of anion induced supramolecular polymer in solution phase. In their experiments they have observed a decrease in diffusion coefficient value with the increasing concentration of the receptor, which gives evidence in support of supramolecular polymer formation in the solution phase. Following the above-mentioned literature reports, we have also performed the 2D-DOSY NMR experiment for the free and anionic complex of the receptor in CHCl₃ solution. At first 2D-DOSY NMR spectrum was recorded for the 5.0 mM solution of the free bis(indole) compound **1c**, which provided the diffusion coefficient value of $5.96 (\pm 0.03) \times 10^{-6} \text{ cm}^2/\text{s}$ (Fig. S28). With the addition of 3.0 equivalent TBACl salt in the same 5.0 mM receptor solution leads the decrement of diffusion coefficient value, which is found to be $4.25 (\pm 0.02) \times 10^{-6} \text{ cm}^2/\text{s}$ (Fig. S29). This decrease in the diffusion coefficient value supports the polymer structure formation in the solution phase, which has formed by the conversion of the double helix form to monomeric unit and subsequent formation of polymeric aggregates.

Further, the 2D-DOSY NMR spectra were recorded at higher concentrations of receptor with 3.0 equivalent of TBACl. And from these experiments the diffusion coefficient values have been calculated and the values are $3.14 (\pm 0.02) \times 10^{-6} \text{ cm}^2/\text{s}$ and $2.64 (\pm 0.02) \times 10^{-6} \text{ cm}^2/\text{s}$ for 14 mM and 20 mM concentration of receptor (with 3.0 equivalent of TBACl), respectively (Fig. S30 and S31). Overall, in the presence of Cl⁻ ion, the diffusion coefficient values decreases from $4.25 (\pm 0.02) \times 10^{-6}$ and $3.14 (\pm 0.02) \times 10^{-6}$ to $2.64 (\pm 0.02) \times 10^{-6} \text{ cm}^2/\text{s}$ with the increasing concentration of bis(indole) compound **1c**, from 5 and 14 to 20

mM, respectively. These observations suggested that our compound form supramolecular polymer in the presence of Cl^- in the solution phase.

The above-mentioned observations and discussions have been incorporated in the revised version of the manuscript and supplementary information files.

Fig. S28. The 2D DOSY NMR spectrum of bis(indole) compound **1c** (5 mM) in CHCl_3 at 25 °C. Two sub-figures in the bottom sections are the screenshots of the processed data. The diffusion coefficient value for this compound in the above-mentioned condition is $5.96 (\pm 0.03) \times 10^{-6} \text{ cm}^2/\text{s}$.

Fig. S29. The 2D DOSY NMR spectrum of bis(indole) compound **1c** (5 mM) with 3 equiv. of TBACl in CHCl_3 at 25 °C. Two sub-figures in the bottom sections are the screenshots of the processed data. The diffusion coefficient value for this compound in the above-mentioned condition is $4.25 (\pm 0.02) \times 10^{-6} \text{ cm}^2/\text{s}$.

Fig. S30. The 2D DOSY NMR spectrum of bis(indole) compound **1c** (14 mM) with 3 equiv. of TBACl in CHCl_3 at 25 °C. Two sub-figures in the bottom sections are the screenshots of the processed data. The diffusion coefficient value for this compound in the above-mentioned condition is $3.14 (\pm 0.02) \times 10^{-6} \text{ cm}^2/\text{s}$.

Fig. S31. The 2D DOSY NMR spectrum of bis(indole) compound **1c** (20 mM) with 3 equiv. of TBACl in CHCl₃ at 25 °C. Two sub-figures in the bottom sections are the screenshots of the processed data. The diffusion coefficient value for this compound in the above-mentioned condition is $2.64 (\pm 0.02) \times 10^{-6} \text{ cm}^2/\text{s}$.

Question 2: While anion transport was evidenced by the HPTS assay and channel formation supported by the observation of single channel current recording, the bundled ion channel structure in the membrane as inferred from crystal structures and schematically shown in Fig. 4b lacks experimental evidence. In the solid-state and perhaps in CHCl₃ or CH₂Cl₂ solutions, supramolecular polymerization leading to channel-like assemblies was driven by Cl⁻ binding which linked adjacent isophthalimides into sheets. However, in lipid bilayer membranes in water, Cl⁻ binding is extremely weak as Cl⁻ is strongly hydrated in water. Therefore, it is unlikely that the same structures observed in crystals would form in lipid bilayers. Single column channels, or other unidentified assembly modes could be responsible for the anion conductance observed. I appreciate the difficulty of studying molecular self-assembly in the membrane - if the authors could not provide direct evidence of the structure of ion channels in the membrane, it would be prudent to re-draw Fig. 4, perhaps only representing single-column channels. The statement of “anion-induced self-assembled supramolecular ion channel formation across the lipid bilayer membrane” should be removed as the channel formation is likely anion independent due to anion binding being extremely weak in water.

Response: We thank the reviewer for the comment. The bis(indole) molecules form stable double helix structure in absence of anion. However, in the presence of anion, the double helix structure breaks apart for the formation of supramolecular channel in polar aprotic solvents. We understand the concern of the reviewer regarding the anion-induced supramolecular channel formation in the aqueous solution. However anion-induced supramolecular channel is already reported in the literature by Prof. Barboiu (suggested by other reviewer), which forms a channel structure in the hydrophobic bilayer membrane and is

found to transport ions across the bilayer membrane (Zheng, S.-P., Jiang, J.-J., van der Lee, A. & Barboiu, M. *Angew. Chem., Int. Ed.* **59**, 18920-18926 (2020)).

To avoid any confusion regarding the channel-forming model inside the bilayer membrane by the bis(indole) molecules, we have synthesized a new derivative **1e** by preplacing the H_h proton by 4-(*tert*-butyl)phenyl group (Scheme S6). We have already discussed from the crystal structure of the anionic complex that the H_h proton plays a crucial role by interacting with the anion for the formation of supramolecular channel. This interaction is reflected in the NMR titration experiment (Fig. 3g and S22, where chemical shift change has been observed) and also in the ion transport experiment across vesicular and planar bilayer membrane. However, the NMR titration experiment of new bis(indole) compound **1e** does not reflect any interaction with the anion, as we have not observed any significant chemical shift change of acidic protons (Fig. S26). This result is supported by the fact that after replacing the H_h proton, the binding energy from other proton-anion interactions is not sufficient enough to break the double helix structure. Therefore, even in the presence of anion, the bis(indole) molecule stayed in the double helix conformation. Furthermore, the above-mentioned experimental observation is supported by the ion transport activity of the new bis(indole) derivative, where we have not observed any significant ion transport activity by the bis(indole) compound **1e** even at 50 times concentrated ($0.5 \mu\text{M}$) solution (Fig. S36).

Fig. S45. Schematic representation of supramolecular polymer formation involving the H_h proton.

Scheme S6. The synthesis scheme of H_b proton substituted bis(indole) derivative **1e**.

Fig. S26. The stacked ¹H NMR spectra of **1e** (3.0 mM) in acetonitrile-*d*₃ at room temperature upon titrating increasing equivalents of TBACl.

Fig. S36. The ion transport activity of bis(indole) compound **1e** (0.5 μM) in HPTS assay. No significant ion transport activity has been observed.

From the above-mentioned experimental observations and considering the literature report of the anion-induced supramolecular channel, we believe that our channel-forming model is possibly correct (or close to the accurate), which is supported by various indirect experimental observations.

The above-mentioned literature reference, synthetic details, experimental observations, and discussions have been incorporated into the revised version of the manuscript and supplementary information files.

Question 3: As discussed theoretically by Berezin (*Supramol. Chem.* 2013, 25, 323) and demonstrated experimentally by Gale (*Chem. Commun.* 2021, 57, 3979), the apparent selectivity determined by varying the extraventricular anion in HPTS base pulse assay is opposite to the true selectivity. To address the anion selectivity, the authors could employ modified methods by Gale or use an acid-pulse instead of a base pulse (see *Chem. Commun.* 10.1039/D2CC00144F). Since anion selectivity is not the focus of the manuscript, without the additional experiments, the term “selectivity order” (line 296) can be replaced by “pH dissipation trends in the order”. In addition, anions like acetate and fluoride should not be used in HPTS assays because the conjugate acids of these anions are membrane permeable and exist in significant amounts at pH 7. The HOAc simple diffusion from outside to the inside of liposomes opposed the direction of H⁺ efflux along the pH gradient, leading to the weakest fluorescence change when OAc⁻ was used as the extraventricular anion (Fig. 5c). This originates from an acid diffusion process and does not provide any information of OAc⁻ selectivity.

Response: We thank the reviewer for the valuable suggestion. Based on the suggestion of the reviewer we have gone through the suggested literature reports ((a) Berezin, S. K. *Supramol. Chem.* **25**, 323-334 (2013); (b) Wu, X. & Gale, P. A. *Chem. Commun.* **57**, 3979-3982 (2021)), where they have discussed the limitations of the anion selectivity assay. From the

above-mentioned literature report, we found that the process followed by us for the anion selectivity assay mentioned in this manuscript is not accurate. However, based on the suggestion of the reviewer we have replaced the “selectivity order” term by “pH dissipation trends in the order” to correlate the discussion part with the experimental observation.

Moreover, we have removed the OAc^- anion from the anion selectivity experiments, as the conjugate acids of these anions (OAc^- and F^-) are membrane permeable and exist in significant amounts at pH 7.

The above-mentioned modifications have been incorporated into the revised version of the manuscript and supplementary information files.

Question 4: Page S37, "planner" should be "planar".

Response: We thank the reviewer for the comment. The “planner” word has been replaced by “planar” in the revised version of the supplementary information file.

REVIEWERS' COMMENTS

Reviewer #2 (Remarks to the Author):

I am satisfied with the answers to my comments and the changes made by the authors to the manuscript. I recommend publication.

Reviewer #3 (Remarks to the Author):

The authors have satisfactorily addressed my concerns regarding self-assembly in solution. However the new derivative 1e does not seem to be a suitable negative control for the anion transport function/channel formation in lipid bilayers as 1e is significantly more lipophilic than 1b and thus won't be delivered into the lipid bilayers as efficiently as 1b. My final recommendation is to use more cautious terms when describing the anion transport/channel assembly in lipid bilayers. For example, in the caption of Fig.4, "The schematic representation" can be changed to "The proposed schematic representation".

Reply to Reviewers' Comment

Comments to the Author

Reviewer #2 (Remarks to the Author):

I am satisfied with the answers to my comments and the changes made by the authors to the manuscript. I recommend publication.

Response: We are highly thankful to the reviewer for giving his/her precious time and carefully reading our work for the evaluation of the manuscript.

Reviewer #3 (Remarks to the Author):

The authors have satisfactorily addressed my concerns regarding self-assembly in solution. However the new derivative **1e** does not seem to be a suitable negative control for the anion transport function/channel formation in lipid bilayers as **1e** is significantly more lipophilic than **1b** and thus won't be delivered into the lipid bilayers as efficiently as **1b**. My final recommendation is to use more cautious terms when describing the anion transport/channel assembly in lipid bilayers. For example, in the caption of Fig.4, "The schematic representation" can be changed to "The proposed schematic representation".

Response: We thank the reviewer for the comment. We agree with the reviewer that the high lipophilicity of compound **1e** might be a reason for its poor permeation into the lipid bilayer membrane and hence no transport was observed. However, when we carried out ¹H NMR titration of **1e** with TBACl (up to 5 equivalent), no chemical shift change was observed for any proton of the molecule (Supplementary Figure 32) i.e., **1e** exists as a double helix even in the presence of Cl⁻, and no supramolecular polymeric structure, similar to **1b**, is formed. The stability of **1e** double helix structure even in the presence of Cl⁻ indicates that the presence of a steric group at H_b proton position completely stops the formation of either chloride-bound supramolecular polymer (similar to **1b**) or any other chloride-bound complex (Supplementary Figure 51). Therefore, the transport of chloride via single column channels or other unidentified assembly modes is unlikely.

However, based on the suggestion of the reviewer, we have modified the channel formation descriptions as: "The proposed schematic representation of the formation of chloride channel inside the lipid bilayer membrane from a Cl⁻ ion bound supramolecular assembly” in Figure 4.

Supplementary Figure 32. The ^1H NMR titration experiment of 1e. The stacked plot of ^1H NMR spectra of 1e (3.0 mM) in acetonitrile- d_3 at room temperature upon titrating increasing equivalents of TBACl.

Supplementary Figure 51. The Polymer formation strategy. Proposed schematic representation of supramolecular polymer formation involving the H_h proton.